

Atmospheric
Chemistry
and Physics

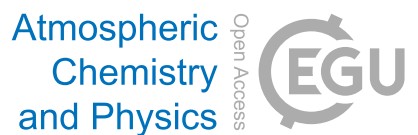

Research article

# Evaluating tropospheric nitrogen dioxide in UKCA using OMI satellite retrievals over south and east Asia

**Alok K. Pandey**[1,2], **David S. Stevenson**[1], **Alcide Zhao**[1], **Richard J. Pope**[3,4], **Ryan Hossaini**[5], **Krishan Kumar**[6], **and Martyn P. Chipperfield**[3,4]

[1]School of GeoSciences, University of Edinburgh, Edinburgh, EH9 3FF, UK
[2]Department of Environmental Sciences, Deshbandhu College, University of Delhi, New Delhi, India
[3]School of Earth and Environment, University of Leeds, Leeds, LS2 9JT, UK
[4]National Centre for Earth Observation, University of Leeds, Leeds, L22 9JT, UK
[5]Lancaster Environment Centre, Lancaster University, Lancaster, LA1 4YQ, UK
[6]School of Environmental Sciences, Jawaharlal Nehru University, New Delhi, India

**Correspondence:** Alok K. Pandey (alok.pandey@ed.ac.uk) and David S. Stevenson
(david.s.stevenson@ed.ac.uk)

**Abstract.** We compare tropospheric column nitrogen dioxide ($NO_2$) in the United Kingdom Chemistry and Aerosol (UKCA) model version 11.0 with satellite measurements from NASA's Earth Observing System (EOS) Aura satellite Ozone Monitoring Instrument (OMI) to investigate the seasonality and trends of tropospheric $NO_2$ over south and east Asia (S and E Asia). UKCA is the atmospheric composition component of the UK Earth System Model (UKESM). UKCA was run with nudged meteorology, producing hourly output over S and E Asia for 2005–2015. OMI averaging kernels have been applied to the model hourly data sampled at Aura's local overpass time of $13{:}45\,\mathrm{LT} \pm 15$ min to allow for consistent model–data comparison. Background UKCA and OMI tropospheric column $NO_2$ typically ranges between $0 \times 10^{15}$ and $2 \times 10^{15}$ molec. cm$^{-2}$. Diurnal cycles and vertical profiles of the tropospheric $NO_2$ column in UKCA show that the daily minimum tropospheric column $NO_2$ occurs around the satellite overpass time. UKCA captures the seasonality but overestimates $NO_2$ by a factor of $\sim 2.5$, especially during winter over eastern China and north India, at times and locations with high aerosol loadings. Heterogeneous chemistry is represented in the version of UKCA used here as uptake of $N_2O_5$ on internally generated sulfate aerosol. However, aerosol surface area may be underestimated in polluted locations, contributing to overestimation of $NO_2$. In addition, the model may underestimate emissions of volatile organic compounds (VOCs) and associated peroxy acetyl nitrate (PAN) formation, leading to insufficient long-range transport of oxidised nitrogen and also contributing to overestimation of $NO_2$ over polluted regions and underestimation over remote regions. Quantifying and understanding discrepancies in modelled $NO_2$ warrant further investigation as they propagate into modelling of multiple environmental issues.

## 1 Introduction

Nitrogen oxides ($NO_x$ is the sum of nitrogen dioxide, $NO_2$, and nitric oxide, NO) are key gases in atmospheric chemistry, and models need to simulate them adequately in order to faithfully represent many important environmental processes. Nitrogen oxides play a central role in the atmospheric nitrogen cycle (Fowler et al., 2013) and are a precursor of nitrate aerosols (Liu et al., 2016) and the greenhouse gas (GHG) tropospheric ozone ($O_3$) (Bucsela et al., 2008; von Schneidemesser et al., 2015). The oxidising capacity of the atmosphere is affected by $NO_x$, so it also influences other GHGs such as methane (Naik et al., 2013; Voulgarakis et al., 2013); hence, changes in $NO_x$ contribute to climate change (Lelieveld et al., 2015). High concentrations of $NO_2$ can

also increase the risk of acute and chronic respiratory diseases (Brunekreef et al., 2009). Deposition of NO$_2$ and other species containing reactive nitrogen can lead to the eutrophication of ecosystems and loss of biodiversity (Stevens et al., 2004; Erisman et al., 2013).

Nitrogen oxides are predominantly emitted as NO, mainly originating from fossil fuel combustion (ca. 58 % of the total), natural emissions (ca. 23 %), and agriculture/biofuel use (ca. 19 %) (Lelieveld et al., 2015). In the sunlit troposphere, NO reacts with O$_3$ to produce NO$_2$, which photolyses to return NO and O$_3$, rapidly forming a photochemical equilibrium. The oxidation products of volatile organic compounds (VOCs) react with NO$_2$ to form peroxy acetyl nitrates (PANs), key constituents of photochemical smog (Sher, 1998; Beirle et al., 2003; Mallik and Lal, 2014). These compounds are stable at low temperatures typical of the upper troposphere but thermally unstable in the lower troposphere, decomposing to release NO$_2$, thus facilitating long-range transport of NO$_2$ from NO$_x$ source regions to remote sites (Fiore et al., 2018). NO$_2$ is mainly removed by dry deposition and via oxidation to nitric acid, which readily deposits. Another sink of reactive oxidised nitrogen (NO$_y$), in darkness, is via heterogeneous uptake of dinitrogen pentoxide (N$_2$O$_5$) on the surface of aerosols (Dentener and Crutzen, 1993), which then deposit. These removal processes typically result in a short lifetime for NO$_2$ of a few hours (e.g. Beirle et al., 2011) and lead to strong spatial and temporal variations in its atmospheric abundance. Atmospheric chemistry models include our best representations of these, and many other, processes that control NO$_x$, allowing models to simulate spatial and temporal variations in atmospheric composition in detail (e.g. Szopa et al., 2021).

Since the 1990s, various satellite-based instruments have measured tropospheric NO$_2$ columns. The Global Ozone Monitoring Experiment (GOME) detected NO$_2$ pollution hotspots around the world (Leue et al., 2001), and in 2002, the scanning imaging absorption spectrometer for atmospheric cartography (SCIAMACHY) began mapping NO$_2$ pollution at a spatial resolution of 30 km × 60 km, with global coverage every 6 d, allowing for the detection of temporal trends in NO$_2$ (van der A et al., 2008). The launch of the Ozone Monitoring Instrument (OMI) in 2004 has provided even higher spatial resolution information (13 km × 24 km) of tropospheric NO$_2$ with daily global coverage (Levelt et al., 2006; Boersma et al., 2008; Liu et al., 2016). OMI NO$_2$ data have been validated against in situ and surface-based observations (e.g. Irie et al., 2009; Lamsal et al., 2014) and provide a long record of high-spatial-resolution daily measurements of NO$_2$, useful for the evaluation of global atmospheric chemistry models (e.g. van Noije et al., 2006).

This study focusses on NO$_2$ pollution over south and east Asia (0–50° N and 55–145° E; Fig. 1a) during the period of 2005–2015. The region is home to nearly 50 % of the Earth's population and has some of the largest measured NO$_2$ columns. Rising energy demand, urbanisation, traffic, and industrialisation have led to increases in NO$_x$ emissions in some regions, whilst technological advances, typically introduced in response to environmental legislation, have led to reductions in NO$_x$ emissions in other locations (Mijling et al., 2013). Regional variations in the evolution of NO$_x$ emissions have been captured by satellite NO$_2$ measurements, with eastern China showing upward trends of tropospheric NO$_2$ up to 2011 followed by decreases since 2012 (Shah et al., 2020; Fan et al., 2021; Cooper et al., 2022). By contrast, India shows a continuous increase of 12.5 % to 29.6 % from 2005 to 2019 (Krotkov et al., 2016; Singh et al., 2023).

We compare model simulations of the NO$_2$ column with equivalent satellite measurements from OMI to evaluate model performance in terms of simulating the magnitude and spatial distribution of NO$_2$ over S and E Asia and its seasonal variations and longer-term temporal trends. We use the UK Chemistry and Aerosol (UKCA) model, the atmospheric composition component of the UK Earth System Model (UKESM). Archibald et al. (2020) compared OMI and UKCA NO$_2$ columns, identifying some model biases that we explore in more detail here. Because of the high reactivity and short atmospheric lifetime of NO$_2$ and its anthropogenic sources, its tropospheric concentration has a distinct diurnal signature as well as a dynamically varying vertical profile. OMI takes column NO$_2$ measurements at a particular time each day (13:45 LT ± 15 min at the Equator). The vertical profile of NO$_2$ at the time of measurement has a strong influence on the column amount measured. This is because OMI measures radiation absorption at specific ultraviolet–visible (UV–vis) wavelengths, sampling air (and hence NO$_2$) along the ray path from the Sun, via the atmosphere, to the satellite; this ray path depends upon the atmospheric albedo (i.e. cloud amount and height) at the time of measurement. An averaging kernel (AK) is required to translate the vertical profile to a column amount; the AK is a weighting profile that depends upon environmental conditions (e.g. cloud properties) at the time of measurement. Model evaluation therefore requires careful temporal sampling and application of the AK so that the model is sampled in the same way as OMI samples the atmosphere. Model simulations nudged by meteorological reanalysis data are used so that the physical state of the model atmosphere resembles the real atmosphere as closely as possible. This study uses a single model (UKCA), but evaluation of NO$_2$ from multiple models (e.g. van Noije et al., 2006), such as in a model intercomparison project (MIP) for oxidised nitrogen, is a desirable aim to extend this work to a wider set of models.

The paper is structured as follows. Section 2 describes the version and experimental setup of the UKCA model used for simulations and the OMI NO$_2$ datasets used to evaluate the model. In Sect. 3, we present model results for diurnal and seasonal variations in the vertical distribution of NO$_2$ and how they influence the NO$_2$ column amount, analysing the spatial distribution over S and E Asia and temporal trends over the period of 2005–2015, comparing model results with

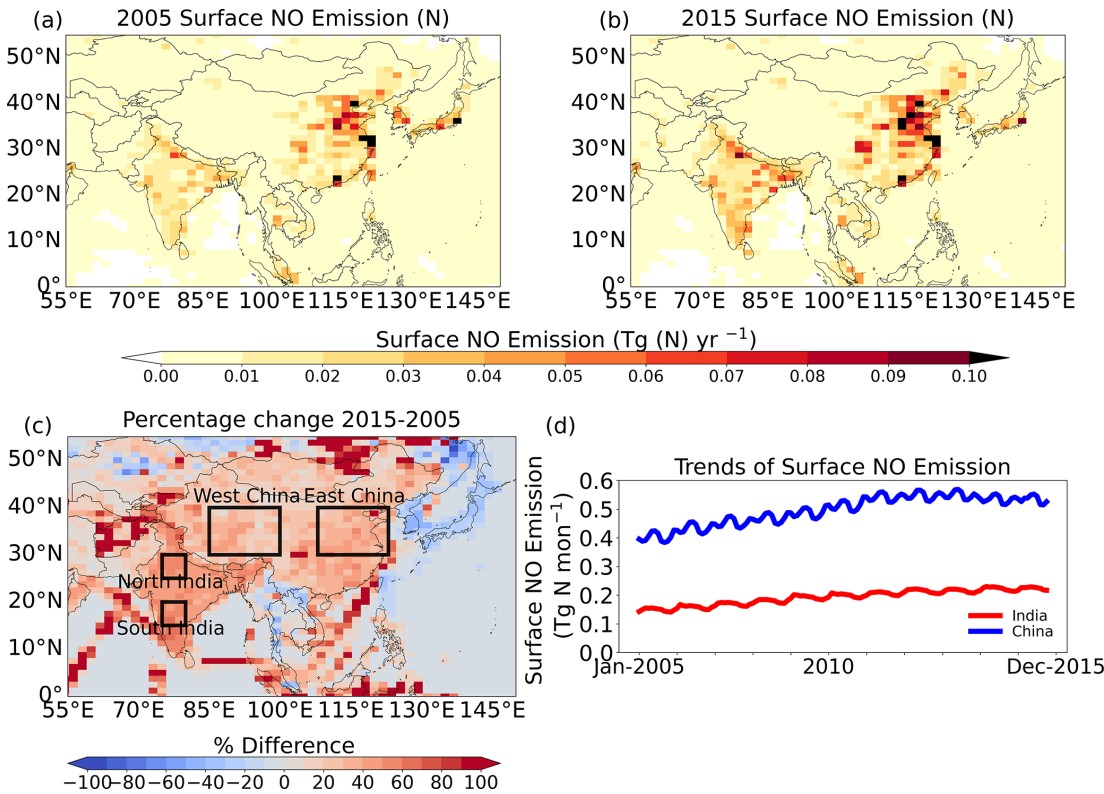

**Figure 1.** Surface nitrogen oxide (NO) emissions over S and E Asia (Tg N yr$^{-1}$) in **(a)** 2005 and **(b)** 2015. **(c)** Percentage change in the NO surface emissions from 2005 to 2015. **(d)** Trends of NO surface emissions (Tg N month$^{-1}$) from 2005 to 2015 over India and China. Boxes shown in panel **(b)** indicate regions referred to in the text.

satellite data. We discuss some of the reasons for model–observation discrepancies in Sect. 4 before drawing conclusions about model performance with respect to representation of NO₂ in Sect. 5.

## 2  Data and methods

We focus our analysis on S and E Asia (Fig. 1a), dividing this region into six different sub-regions, two political (India and China) and four geographical (E China, W China, N India, and S India; Fig. 1c). The surface NO emissions over south Asia and east Asia in 2005 and 2015 are shown in Fig. 1a and b, respectively. We focus on the N India and E China sub-regions for detailed study as these are hotspots of high population density and high NO₂ emissions (Ramachandran et al., 2013; Sekiya et al., 2018). Apart from country-wise analysis, we also selected NO$_x$ hotspot and cleaner regions to capture contrasting pollution profiles across S and E Asia. There are large differences in NO emission intensity (i.e. emissions per unit area) across the region: W China has NO emission intensity approximately 30 times lower than E China, while S India's emissions intensity is relatively low at about half that of N India (Fig. S1 in the Supplement). This approach allows for a comprehensive analysis across both high-emission and cleaner regions, providing broader insights into NO$_x$ distribution. TS1 In 2015, total surface NO emissions were around 0.09–0.10 Tg N yr$^{-1}$ from the box over E China and 0.07–0.08 Tg N yr$^{-1}$ over the N India box. Figure 1c shows percentage changes relative to 2005 in the NO surface emissions from 2005 to 2015 (from AerChemMIP, Collins et al., 2017), which show a 40 %–60 % increase in surface NO emissions over this period for India, whereas the increase is relatively smaller (20 %–40 %) for China. The 2005–2015 trends in surface NO emissions integrated over the whole of China and India are shown in Fig. 1d. Surface NO emissions from China were 4.8 Tg N yr$^{-1}$ in 2005, increasing to 6.5 Tg N yr$^{-1}$ by 2011, followed by a decrease to 5.8 Tg N yr$^{-1}$ in 2015, as also reported in other studies (Miyazaki et al., 2017; Shah et al., 2020). In contrast, India has shown a consistent upward trend, with NO emissions increasing from 1.8 Tg N yr$^{-1}$ in 2005 to 2.5 Tg N yr$^{-1}$ in 2015 (Krotkov et al., 2016).

### 2.1  UKCA model

We use the United Kingdom Chemistry and Aerosols (UKCA) model version 11.0 (Archibald et al., 2020). UKCA is an aerosol–chemistry model coupled with the UK Met Office Hadley Centre HadGEM family of climate models. UKCA simulates the atmospheric composition and climate

from the surface to the mesosphere (Morgenstern et al., 2009). HadGEM acts as the dynamical core and provides components for large-scale advection, convective transport, and boundary layer mixing of chemical and aerosol tracers (O'Connor et al., 2014). UKCA version 11.0 comprises the GA7.1 climate model (Walters et al., 2019) with the StratTrop (CheST) chemistry scheme and the GLOMAP-mode aerosol scheme. The UKCA stratospheric and tropospheric chemistry schemes include all the well-known photochemical and nighttime reactions related to NO$_x$ and are described in detail and evaluated by Morgenstern et al. (2009), O'Connor et al. (2014), and Archibald et al. (2020). Aerosol surface area from GLOMAP is used to drive heterogeneous chemistry.

The model's horizontal resolution (N96, 1.875° longitude × 1.25° latitude) is much coarser than the satellite data products used. The model is divided into 85 hybrid height levels with the model top at ∼ 85 km. The vertical resolution is the finest close to the surface and gradually decreases with height; i.e. layers are concentrated towards the surface, so the boundary layer (BL) is relatively well resolved in the model, with the lowest (surface) level being ∼ 18 m thick.

We used a variant of the version 11.0 "release job" (job ID u-bb210; https://www.ukca.ac.uk/wiki/index.php/Release_Job_UM11.0, last access: 24 April 2025), adding a meteorological nudging scheme to allow for a more meaningful comparison of satellite data to model output. Nudging (Newtonian relaxation) is a data assimilation technique that adjusts dynamical variables of a free-running general circulation model (GCM) using meteorological reanalysis data to allow for a relatively realistic representation of the atmosphere at a given time. For nudging, the European Centre for Medium-range Weather Forecasts (ECMWF) ERA-Interim data were used, and the model was run from 2005 to 2015. The ERA-Interim data are at T255 (78 km) resolution on hybrid $p$ levels, provided at 6 h intervals. These variables are then interpolated to the model's N96 resolution.

Monthly varying Coupled Model Intercomparison Project Phase 6 (CMIP6) anthropogenic and biomass burning emissions of NO$_x$ and other relevant species from AerChemMIP have been used (Collins et al., 2017). No diurnal variations in anthropogenic or biomass burning emissions are applied. Natural emissions are as described by Archibald et al. (2020); in particular, lightning NO$_x$ emissions are interactive and follow the Price and Rind (1992) parameterisation, whilst soil NO$_x$ emissions vary monthly but are annually invariant and use the Yienger and Levy (1995) distribution.

Archibald et al. (2020) describe the dry and wet deposition schemes, with their Table 1 listing all oxidised nitrogen species deposited in the model. Dry deposition follows a resistance in series approach (Wesely, 1989), with surface resistances assigned according to the surface types specified by the JULES land surface model (Harper et al., 2018). Wet deposition is calculated using a first-order removal scheme driven by the three-dimensional distribution of convective and stratiform precipitation (Giannakopoulos et al., 1999) and scavenging coefficients related to Henry's law coefficients for each species.

## 2.2 Satellite NO$_2$ data

The Ozone Monitoring Instrument (OMI) is a nadir-viewing sensor that measures radiation at ultraviolet–visible wavelengths, mounted on NASA's EOS Aura satellite (Boersma et al., 2011; Liu et al., 2016). Aura travels at an altitude of 705 km in a Sun-synchronous polar orbit and provides daily global coverage with a daytime local Equator crossing time of 13:45 LT ± 15 min (Shah et al., 2020). OMI measures backscattered radiation from the Earth's atmosphere and surface over the wavelength range of 264–504 nm, with a spectral resolution between 0.42 and 0.63 nm and a nadir spatial resolution of 13 km × 24 km (Dobber et al., 2006; Levelt et al., 2006). The instrument consists of a telescopic system using CCD detectors which provide it a 114° field of view, corresponding to a large swath of 2600 km at the Earth's surface. OMI retrieves the ozone column and profile; aerosols; SO$_2$; NO$_2$; and other trace atmospheric constituents such as HCHO, BrO, and OClO using the technique of differential optical absorption spectroscopy (DOAS). OMI tropospheric column NO$_2$ data utilised here come from the Tropospheric Emission Monitoring Internet Service (TEMIS) product (Boersma et al., 2011) (DOMINO v2.0). The data have been screened to only include data with a cloud fraction of below 0.2 in addition to the good data flags while excluding data with the OMI row anomaly using the algorithm of Duncan et al. (2013). This product includes AK information which have been used for model–satellite comparison.

The AK describes the vertical structure of the atmospheric profile, accounting for the measurement sensitivity at different locations and times (Vijayaraghavan et al., 2008; Boersma et al., 2016). In other words, the AK is a linear representation of the vertical weighting of information content of retrieval parameters. The AK is specified as a vector used to provide a measure of the vertical resolution of the estimate (Martin, 2008; Vijayaraghavan et al., 2008). The AKs have been applied to the UKCA model NO$_2$ as shown in Eq. (1):

$$y = \boldsymbol{A} \cdot \boldsymbol{x}, \tag{1}$$

where $\boldsymbol{A}$ is the tropospheric AK, from the OMI product, with vertical values at specific pressures; $\boldsymbol{x}$ is the model profile (sub-columns in units of molec. cm$^{-2}$), interpolated to the OMI vertical pressure grid; and $y$ is the modified model tropospheric column (model sub-columns with AKs applied totalled up to the satellite-defined tropopause). The AKs of each day have been applied to daily model profiles, which are then averaged to produce monthly means. This modified column is then directly compared to the satellite NO$_2$ column.

In addition to applying the AK, the model data must be sampled at the satellite overpass time. We achieve this by

producing hourly model output and matching this to the satellite data. To understand the impacts of sampling at 13:45 LT $\pm$ 15 min, we compare monthly average NO$_2$ values (i.e. an average across all times of day) with a monthly average calculated just using values for between 13:00 and 14:00 LT. To account for the resolution difference, the OMI satellite data were spatially averaged to match the coarser resolution of the UKCA model grid (N96). This ensures consistency between the datasets and allows for a fair comparison by aligning the spatial scales of observations and simulations. For model–observation comparisons, only days and grid boxes with valid satellite retrievals (cloud fraction < 0.2) are included. This ensures consistency in spatial and temporal sampling, though the comparison may be affected by the clear-sky bias inherent in satellite observations as NO$_2$ columns under cloudy conditions are excluded from the analysis.

We used linear regression to calculate trends in tropospheric NO$_2$ concentrations for both model (UKCA) and observation (OMI) datasets. Annual means were used, inherently removing seasonal variability. Statistical significance was tested at a 95 % confidence level ($\alpha = 0.05$). The $t$ statistic for each trend was calculated as $t = \text{trend}/\text{SE}$, where SE is the standard error in the trend estimate. The critical $t$ value ($t_{\text{critical}}$) was obtained from the $t$ distribution using degrees of freedom (df $= n - 2$, where $n$ is the number of years). Trends were considered significant if $|t| > t_{\text{critical}}$, indicating that the trend is unlikely to have occurred by chance with 95 % confidence. This approach provides a spatially resolved representation of NO$_2$ trends and their statistical reliability.

## 3 Results and discussion

### 3.1 Seasonal and diurnal variations in the vertical profile of NO$_2$

Figure 2 shows seasonal and diurnal variations in tropospheric column NO$_2$ taken directly from the UKCA model (with no vertical weighting) over N India and E China. UKCA tropospheric column NO$_2$ is generally lower over both regions during the late morning and early afternoon (the satellite overpass time) due to photochemical destruction which peaks around local mid-day before NO$_2$ increases in the late afternoon. While NO$_2$ levels remain high during the evening and night in E China, column values decrease in N India. The diurnal cycle shows the lowest daily range in June–July–August (JJA), varying from $\sim 5 \times 10^{15}$–$10 \times 10^{15}$ molec. cm$^{-2}$ (N India) and $\sim 9 \times 10^{15}$–$11 \times 10^{15}$ molec. cm$^{-2}$ (E China). By contrast, the December–January–February (DJF) diurnal cycle shows the largest ranges: $8 \times 10^{15}$–$17 \times 10^{15}$ molec. cm$^{-2}$ (N India) and $30 \times 10^{15}$–$55 \times 10^{15}$ molec. cm$^{-2}$ (E China). Please note that the diurnal variations depicted are solely from model simulations and cannot be directly compared with OMI data as there is only one observation time per day. Seasonal diurnal vari-

ations in tropospheric column NO$_2$ for all sub-regions are shown in Fig. S2. It is important to note that the UKCA does not have a diurnal cycle in emissions, so the model does not simulate higher NO$_2$ levels related to real-world processes like late-afternoon rush hour. Rather, higher levels of NO$_2$ in the late afternoon arise solely due to dynamical and photochemical processes.

Figure 3 shows the seasonal variation in vertical profiles of NO$_2$ over N India and E China as simulated by the UKCA model. The highest levels of NO$_2$ are found in the boundary layer, close to sources, and during winter, when the boundary layer is the shallowest and when NO$_2$ loss chemistry proceeds more slowly. Levels of NO$_2$ above the boundary layer are much lower and show seasonal maxima in summer at 10 km. NO$_2$ in the upper troposphere reflects a balance between sources associated with enhanced convection and lightning during the monsoon being partly offset by the higher summer photolysis rates. Equivalent average seasonal vertical profiles (2005–2015) for all regions are shown in Fig. S3, whereas Fig. S4 shows the trends of the vertical profiles from 2005 to 2015 over all regions, which highlights that the vertical extent of NO$_2$ is relatively less affected by pollution in W China and S India in comparison to E China and N India. In addition to pollution levels, meteorological factors, particularly temperature, play a significant role in modulating the vertical distribution and lifetime of NO$_2$. Lower winter temperatures slow chemical reactions, extending the NO$_2$ lifetime. This effect, combined with shallow boundary layers and stable atmospheric conditions, contributes to higher NO$_2$ concentrations near the surface and alters its vertical distribution (Atkinson, 2000; Liu et al., 2016).

Figure 4 shows the diurnal variation in the NO$_2$ vertical profiles over the same regions for the four seasons. The solid black line in Fig. 4 shows the boundary layer height (BLH) of the model, which is highest during the afternoon ($\sim 1$–2 km). Higher surface NO$_2$ values occur at night, and the overpass time of OMI is close to the daily minimum values of NO$_2$ throughout the vertical column and the maximum BLH. Typically, an increase in the surface NO$_2$ concentration is observed after sunset, and the modelled BLH rapidly collapses to well below 100 m. Comparative vertical profiles for all regions are shown in Fig. S5. Diurnal and seasonal variations in the boundary layer height for UKCA, ERA5, and ERA-Interim are shown in Figs. S6 and S7. These variations significantly affect NO$_2$ vertical profiles, particularly during the night, concentrating NO$_2$ near the surface.

### 3.2 Model–satellite data comparisons: time sampling and averaging kernel impacts

The importance of time sampling and application of the AK for model–satellite comparison is shown in Fig. 5 using monthly mean OMI data averaged over the whole of S and E Asia for 2005 and comparing it with UKCA column NO$_2$ data generated in several ways: (i) simple monthly mean,

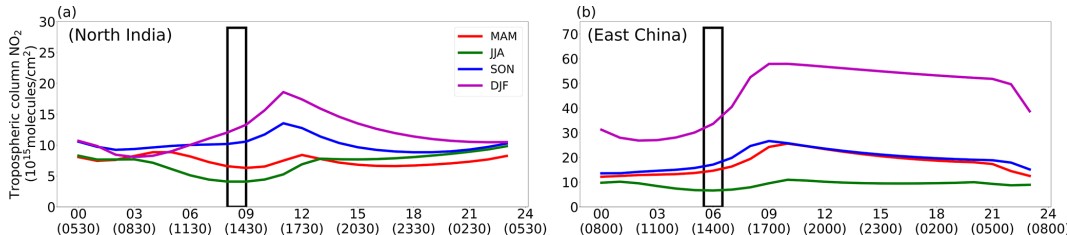

**Figure 2.** Diurnal cycles of tropospheric column $NO_2$ ($10^{15}$ molec. cm$^{-2}$) simulated by UKCA over **(a)** north India and **(b)** east China for the four seasons (averaged over 2005–2015). The time axis displays UTC times, with local time shown in parentheses. The box highlights the OMI overpass time, representing the period used for UKCA-OMI comparisons.

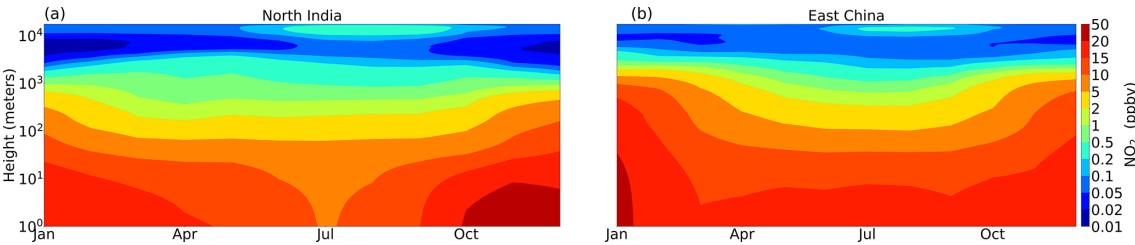

**Figure 3.** Average seasonal vertical profiles (2005–2015) of $NO_2$ (ppbv) in UKCA over **(a)** north India and **(b)** east China.

with no AK weighting; (ii) time-matched to the satellite overpass time using hourly model data but with no AK weighting; and (iii) time-matched and modified by the AK weighting. Measurement uncertainty is based on the daily variation over the month.

Figure 5 shows S and E Asia regional mean OMI tropospheric column $NO_2$ ranges between $1.0 \times 10^{15}$ and $2.0 \times 10^{15}$ molec. cm$^{-2}$ over 2005, with measurement uncertainty $0.5 \times 10^{15}$–$1.0 \times 10^{15}$ molec. cm$^{-2}$. In comparison, the UKCA simple monthly mean tropospheric column $NO_2$ values are larger: $2.2 \times 10^{15}$–$2.5 \times 10^{15}$ molec. cm$^{-2}$ in summer and over $4.0 \times 10^{15}$ molec. cm$^{-2}$ in winter. Whilst the OMI tropospheric column $NO_2$ is measured at 13:45 LT, when the $NO_2$ is typically relatively low (Figs. 2 and 4), the modelled simple monthly mean incorporates all time periods. Therefore, the simple monthly mean modelled $NO_2$ is substantially larger. In contrast, once the diurnal cycle is accounted for (i.e. UKCA is sub-sampled at the satellite overpass time), modelled $NO_2$ is in much better agreement with OMI, with near-zero biases in summer, but in winter, the model still overestimates (by $\sim 80\%$). When the AKs are applied to the model (in addition to sub-sampling at 13:45 LT + 15 min), the summer biases remain near-zero, the winter overestimation is reduced (to $\sim 50\%$), and the model seasonal cycle now sits within the satellite uncertainty range. The inclusion of the AKs has a greater impact in winter due to the shallow boundary layer which confines $NO_2$ near the surface (e.g. as illustrated by UKCA in Figs. 4 and S5), where satellites like OMI are less sensitive because of increased aerosols, clouds, and reduced sunlight. These factors require stronger AK corrections to align models with satellite data. Boersma et al.

(2008, 2016) emphasise the importance of AKs in reducing such discrepancies, while Martin (2008) highlights their role in adjusting for seasonal and vertical variability in $NO_2$ profiles.

Figure 5 clearly demonstrates the importance of accounting for satellite vertical sensitivities and temporal sampling when evaluating model simulations. The variation in seasonal biases following the use of the AKs is more pronounced in winter and less evident in summer. Applying the correct time sampling is much more important than including the AK effect. In all the subsequent analysis presented here, we only show UKCA $NO_2$ columns sampled at the overpass time and with the AK applied.

## 3.3 Seasonal and spatial variations in tropospheric $NO_2$ column

Figure 6 shows the seasonal distribution of tropospheric $NO_2$ observed by OMI and simulated by UKCA, averaged between 2005 and 2015 over S and E Asia. The largest tropospheric $NO_2$ columns can be seen over E China in DJF from both OMI ($> 20 \times 10^{15}$ molec. cm$^{-2}$) and UKCA ($> 30 \times 10^{15}$ molec. cm$^{-2}$). The seasonal minimum (in JJA) tropospheric column values compare well between OMI and UKCA and typically peak around $6 \times 10^{15}$–$10 \times 10^{15}$ molec. cm$^{-2}$. Comparing UKCA and OMI indicates that the model is overestimating tropospheric column $NO_2$ in the major polluted regions (e.g. E China and the Indo-Gangetic Plain), especially in DJF. Over E China the differences range from $+50\%$ to $+100\%$ in March–April–May (MAM), JJA, and September–October–November (SON). In DJF, the model overestimation (over $+150\%$) is more

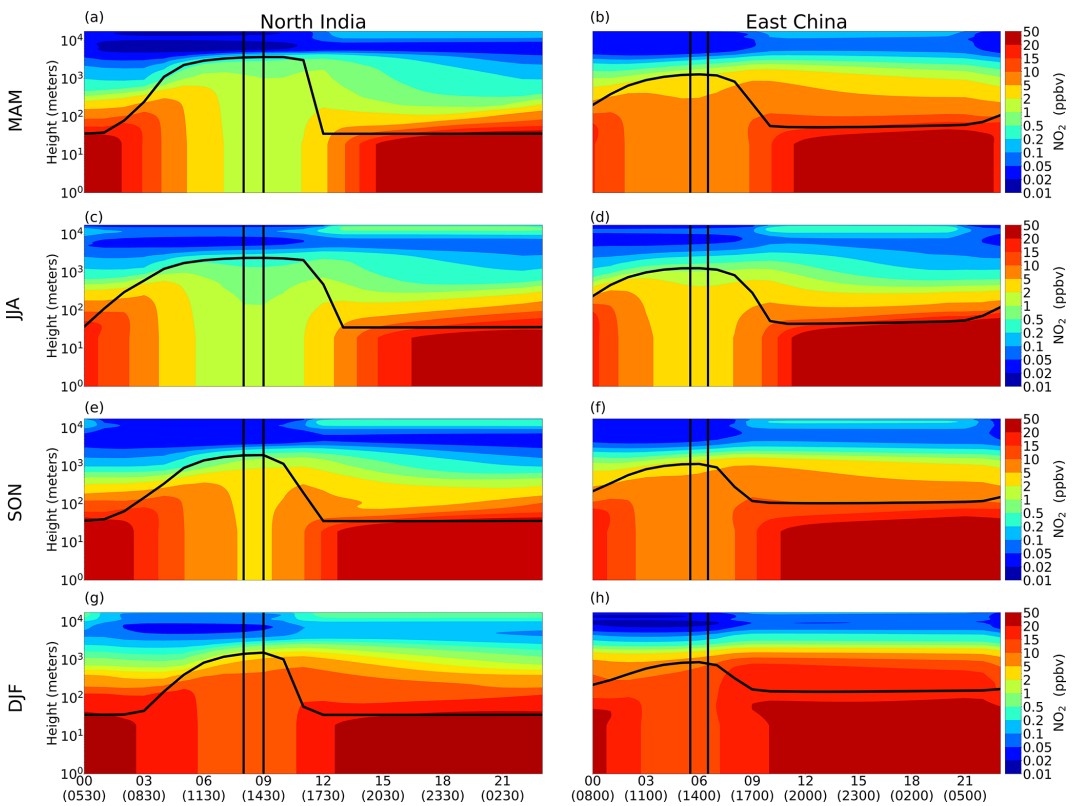

**Figure 4.** Diurnal vertical profile of NO₂ (ppbv) simulated by UKCA over **(a, c, e, g)** N India and **(b, d, f, h)** E China for the four seasons (averaged over 2005–2015). The time axis shows the time in UTC and, in brackets, the local time. The box is the OMI overpass time. The solid black line shows the boundary layer height in the UKCA model.

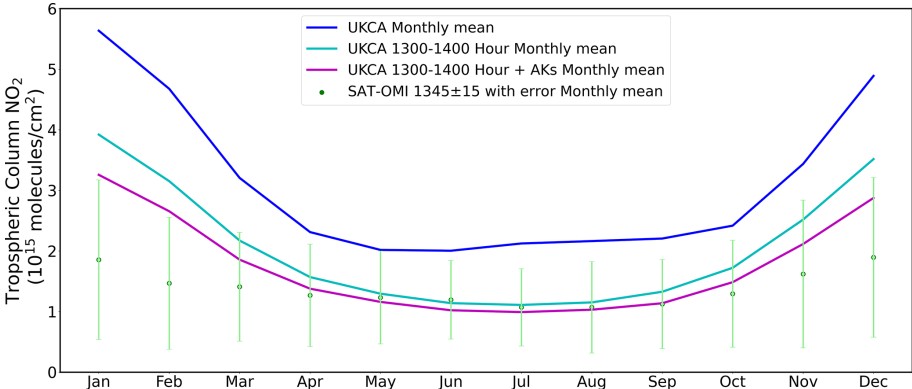

**Figure 5.** Comparison of monthly mean tropospheric column NO₂, averaged over 2005–2015 for the whole S and E Asia region (Fig. 1) from OMI (green, with uncertainty indicated by shading), and from UKCA sampled in three different ways: (i) simple monthly mean (blue); (ii) sampled at the OMI overpass time (cyan); and (iii) sampled at the overpass time and with satellite averaging kernels applied (magenta).

widespread and also covers the pollution outflow regions (e.g. Pacific Ocean). The peak biases in India are also in DJF and are from +100 % to +150 %. In the background, less polluted regions, the model tends to underestimate the observations by up to 100 % in all seasons.

Scatter plots of OMI vs. UKCA tropospheric column NO₂ (Fig. 7) confirm that UKCA overestimates observations in polluted regions in all seasons. The model generally performs the best in JJA and the worst in DJF. UKCA captures the observed spatial variability well, with $R^2$ values of 0.87, 0.77, 0.89, and 0.88 for MAM, JJA, SON, and DJF, respectively. To understand the biases due to the higher values, another best fit, after removing the highest 10 % of observed values, has been computed and plotted (green line).

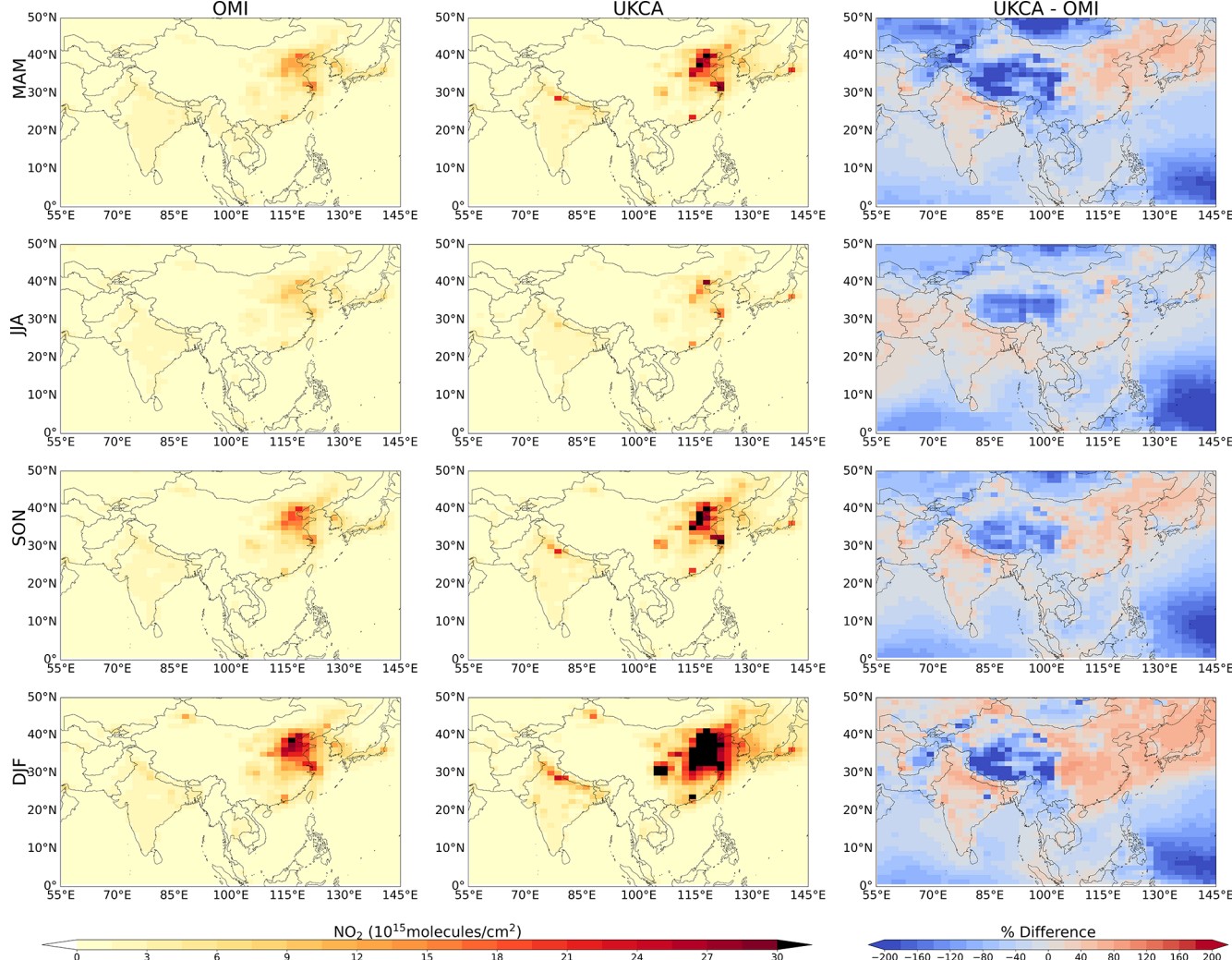

**Figure 6.** Seasonal tropospheric column NO$_2$ ($10^{15}$ molec. cm$^{-2}$) distributions from OMI (left), simulated by UKCA (middle), and the percentage difference ($100\% \times$ (UKCA-OMI)/UKCA) between UKCA and OMI (right).

This shows that the model is performing better over the first 90 % of the OMI data although the fit is not improved in DJF. The main problem with the model appears to be an overestimate of NO$_2$ column over the most polluted regions, especially in winter. Seasonal variations in the boundary layer height (Fig. S7) reveal discrepancies between UKCA and ERA datasets, which may partly explain the model's overestimation of NO$_2$ columns during winter.

## 3.4   Regional OMI and UKCA tropospheric column NO$_2$ variability

Figure 8 shows time series (2005–2015) of OMI and UKCA simulated tropospheric NO$_2$ over the whole of India and China together with over the four regions indicated by the boxes in Fig. 1b. OMI tropospheric column NO$_2$ typically varies between $1 \times 10^{15}$ and $2 \times 10^{15}$ molec. cm$^{-2}$ over India (Fig. 8a) with a relatively small seasonal cy-

cle. Over China, tropospheric column NO$_2$ ranges between $2 \times 10^{15}$ and $4 \times 10^{15}$ molec. cm$^{-2}$ (Fig. 8b) with more pronounced seasonality. UKCA tropospheric column NO$_2$ typically ranges between $2 \times 10^{15}$ and $5 \times 10^{15}$ and $2 \times 10^{15}$ and $12 \times 10^{15}$ molec. cm$^{-2}$ over India and China, respectively. Seasonality is captured by UKCA, but the amplitude is overstated by a factor of 2–3.

Figure 8c and d show UKCA and OMI tropospheric column NO$_2$ over N India and E China, where values typically range between $1 \times 10^{15}$ and $2 \times 10^{15}$ and $5 \times 10^{15}$ and $20 \times 10^{15}$ molec. cm$^{-2}$, respectively. Rapid industrialisation, urbanisation, and increased traffic activity have resulted in a significant increase in the air pollution over E China and N India in the past few decades (Ghude et al., 2008; Kar et al., 2010; Mijling et al., 2013). This can be seen in the OMI data in E China between 2005 and 2011 as tropospheric column NO$_2$ has increased from approximately $12 \times 10^{15}$

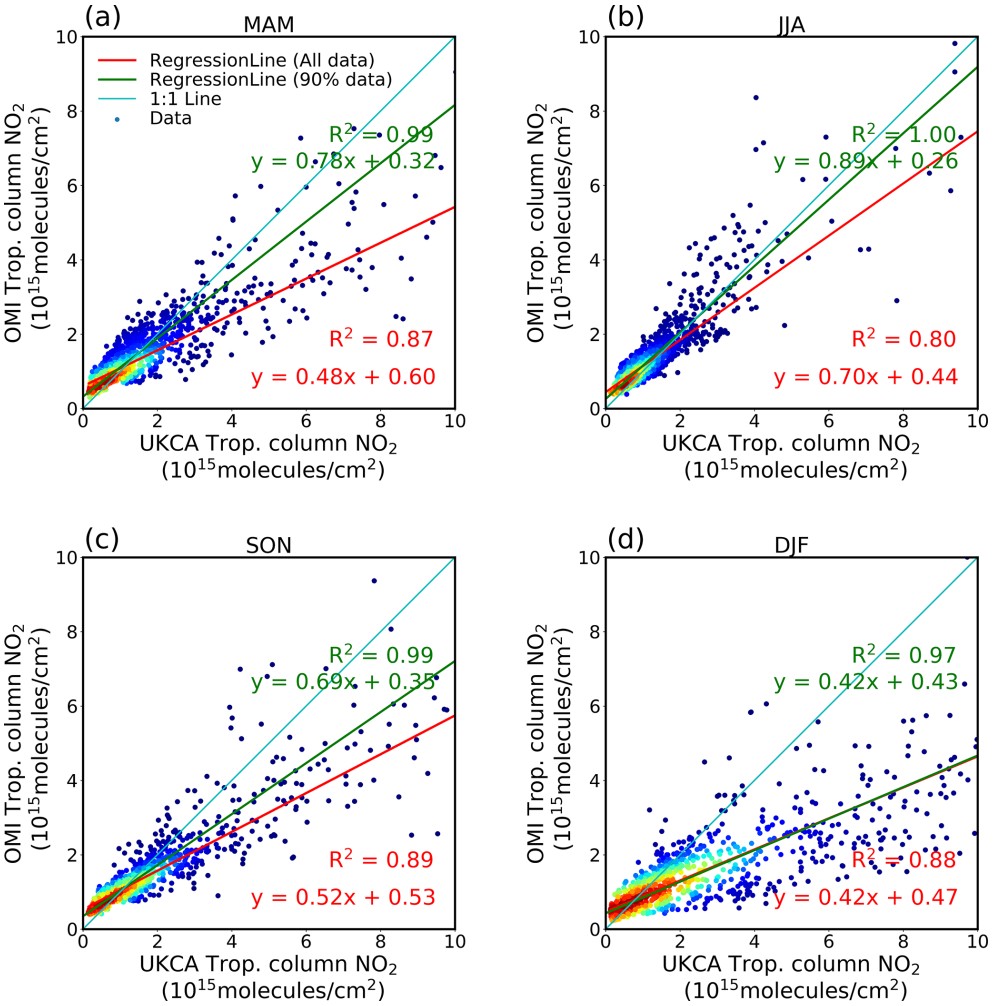

**Figure 7.** Scatter plots of OMI and UKCA tropospheric column NO$_2$ for the four seasons averaged over 2005–2015. Scatter data points are plotted as a heat map, where red corresponds to more data. The 1 : 1 line is shown in cyan colour and best fit by the red line (all data) and green line (lowest 90 % of data). The equations of best fit and the coefficients of determination ($R^2$) are also shown in the respective colours.

to $19 \times 10^{15}$ molec. cm$^{-2}$. The signal in N India is much smaller. Again, UKCA tropospheric column NO$_2$ captures the observed seasonality ($5 \times 10^{15}$–$12 \times 10^{15}$ molec. cm$^{-2}$, N India; $4 \times 10^{15}$–$45 \times 10^{15}$ molec. cm$^{-2}$, E China) but overstates the amplitude. UKCA reproduces the observed trend in E China, but not S India, and overestimates their magnitudes in both cases.

OMI and UKCA trends over the relatively clean regions of S India and W China are shown in Fig. 8e and f, respectively. S India has an OMI lower tropospheric column NO$_2$ ($1 \times 10^{15}$–$2.5 \times 10^{15}$ molec. cm$^{-2}$), and UKCA provides a good representation of the observed seasonality and magnitude. UKCA reproduces the observed marginal increase in tropospheric column NO$_2$ between 2005 and 2011. In W China, the observed tropospheric column NO$_2$ ranges between 0.5 and $1.5 \times 10^{15}$ molec. cm$^{-2}$, which UKCA struggles to reproduce in magnitude ($\sim 50\%$ lower).

## 3.5 Trends of NO$_2$ over the years 2005–2011 and 2011–2015

Over China NO$_x$ emission increased by 52 % from 2005 to 2011 and thereafter decreased by 21 % from 2011 to 2015 (Fig. 1c) as reported elsewhere (De Foy et al., 2016; Liu et al., 2017). Therefore, we focus on OMI and UKCA trends over these time periods; the year 2011 is included in both time periods. Figures 9 and 10 show the spatial distribution of significant NO$_2$ trends for the periods of 2005–2011 and 2011–2015, respectively. We observed the largest trends in DJF, particularly in UKCA, between 2005 and 2011. The seasonal variations in OMI trends are small between 2005 and 2011, with differences of approximately $0.5 \times 10^{15}$ molec. cm$^{-2}$ yr$^{-1}$. Equivalent data showing percentage changes from 2005 to 2011 are shown in Fig. S8. However, there have been increases in DJF across E China of up to $1.0 \times 10^{15}$ molec. cm$^{-2}$ yr$^{-1}$ and decreases of up to

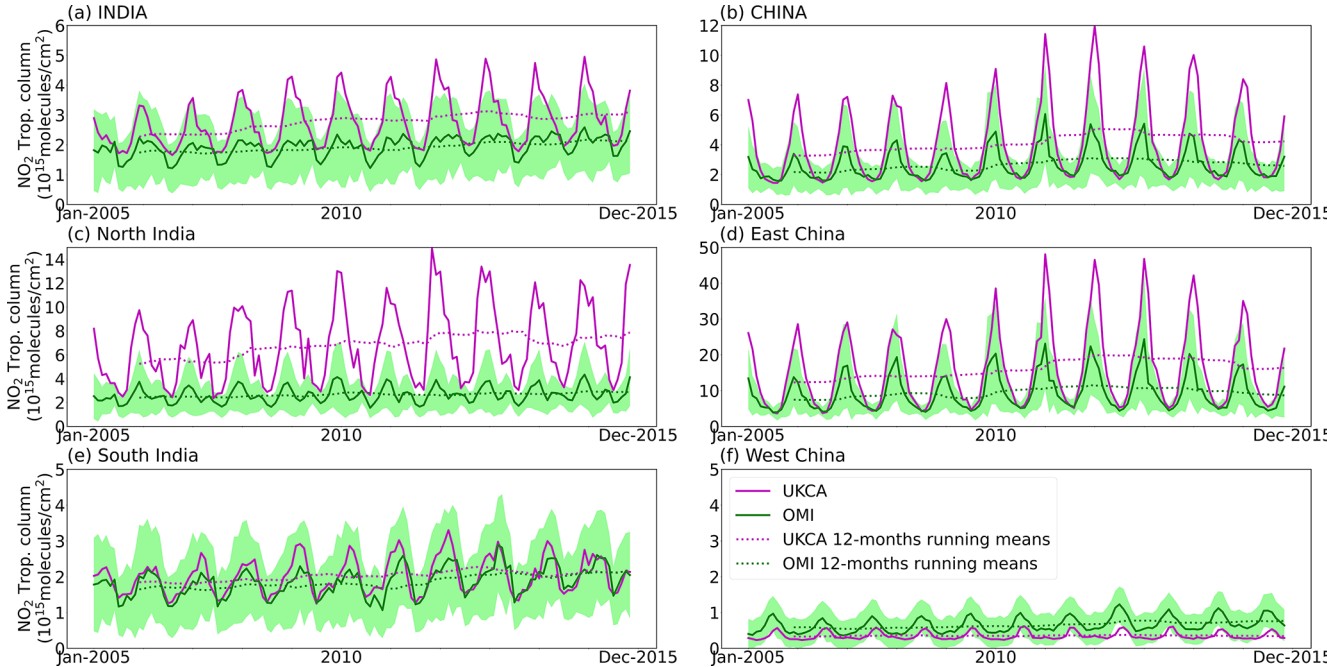

**Figure 8.** OMI and UKCA tropospheric column NO$_2$ ($10^{15}$ molec. cm$^{-2}$) time series over **(a)** India, **(b)** China, **(c)** north India, **(d)** east China, **(e)** south India, and **(f)** west China. The 12-month running means are shown by the dotted lines. Regions are indicated by the boxes in Fig. 1b. Green shading represents the uncertainty in the OMI data.

$0.5 \times 10^{15}$ molec. cm$^{-2}$ yr$^{-1}$ observed over Japan, South Korea and Hong Kong SAR. UKCA shows similar spatial distributions of changes across the majority of the domain, but overstates the magnitudes of decreases over Japan, South Korea, and Hong Kong SAR and increases over E China. There are also substantial model decreases (approximately $-1.0 \times 10^{15}$ molec. cm$^{-2}$ yr$^{-1}$) over east China in SON, which are not present in the OMI observations. Between 2011 and 2015, both OMI and UKCA changes show a steady decrease of up to $2.0 \times 10^{15}$ molec. cm$^{-2}$ yr$^{-1}$ over E China in almost all seasons (Fig. 10 right panel). There are only small changes in the OMI trends over the India from 2011–2015, although a decrease of up to $0.5 \times 10^{15}$ molec. cm$^{-2}$ yr$^{-1}$ is observed in OMI over N India in DJF (Fig. 10 left panel). The corresponding percentage data from 2011 to 2015 is presented in Fig. S9. Significant trends, determined at the 95 % confidence level, reveal distinct spatial and seasonal patterns. For 2005–2011, some regions, especially in winter (DJF), show significant increases, indicating seasonal variations in emissions and boundary layer dynamics (Fig. 9). For 2011–2015, the trends are more pronounced, with notable decreases in NO$_2$ over E China in all seasons, which is consistent with emission reductions during this period (Fig. 10). However, in northeast China, discrepancies between observed and modelled trends suggest uncertainties in the emission inventories used in the UKCA model. The UKCA model captures these significant trends in many regions, though some discrepancies remain, particularly in the

magnitude of the trends. These results highlight the ability of the UKCA model to not only reproduce observed NO$_2$ changes but also underscore areas requiring improvement.

Figures 11 and 12 compare seasonal trends between UKCA and OMI over 2005–2011 and 2011–2015, respectively. UKCA overestimates the magnitudes of trends in NO$_2$ at most locations, with the gradients of best fits (OMI trend over the UKCA trend) in the range of 0.15–0.39 for the 2005–2011 (Fig. 11), but showing a closer correspondence (0.39–0.67) for 2011–2015 (Fig. 12), when the NO$_2$ tropospheric column starts decreasing over China. The overestimation of trends by the model is consistent with the overestimation of NO$_2$ columns in polluted regions, again with the worst agreement in DJF and better performance in JJA.

## 4   Discussion

It is well understood that to usefully compare satellite measurements of column NO$_2$ with model simulations, the model atmosphere needs to be sampled in the same way that the satellite samples the real atmosphere (e.g. Boersma et al., 2008, 2011, 2016). Sampling UKCA at the OMI overpass time and application of a satellite-derived vertical weighting function (averaging kernel) significantly influence the modelled NO$_2$ column and make it more comparable to the OMI values (Fig. 5), although differences remain, particularly during winter, when UKCA overestimates NO$_2$ columns.

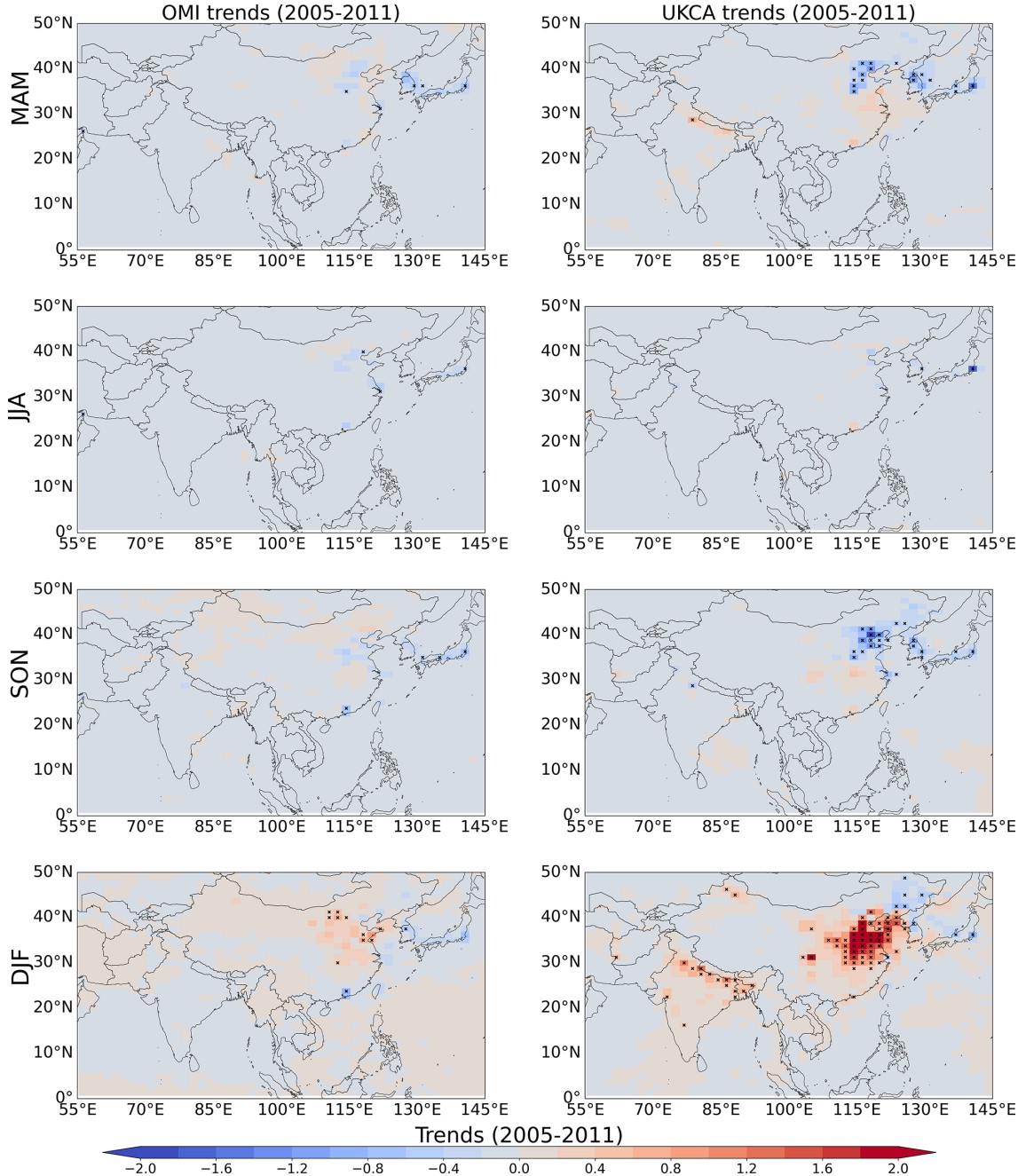

**Figure 9.** Trends of tropospheric column NO$_2$ ($10^{15}$ molec. cm$^{-2}$ yr$^{-1}$) from 2005 to 2011 from OMI (left) and UKCA (right) for the four seasons. Scatter plots of these data are shown in Fig. 11. Equivalent data in percentages are shown in Fig. S8. Crosses indicate grid squares with significant trends.

The results presented in Figs. 2 and 4 illustrate some of the challenges faced by models in accurately simulating column NO$_2$ values measured by satellite instruments such as OMI, particularly during winter (DJF) and at higher latitudes. Diurnal variations in simulated column NO$_2$ for N India and E China (Fig. 2) show that at the OMI overpass time the column is changing the least (it is approximately flat) in JJA, whilst in DJF, it is rising towards a late-afternoon peak,

particularly further north. This means that any errors in the shape of the simulated diurnal cycle of NO$_2$ will translate into larger errors in column NO$_2$ in winter and at higher latitudes.

One source of error in the simulated diurnal cycle of NO$_2$ arises due to the use of diurnally invariant anthropogenic and biomass burning emissions in these UKCA simulations. Boersma et al. (2008) show that using diurnally varying NO$_x$

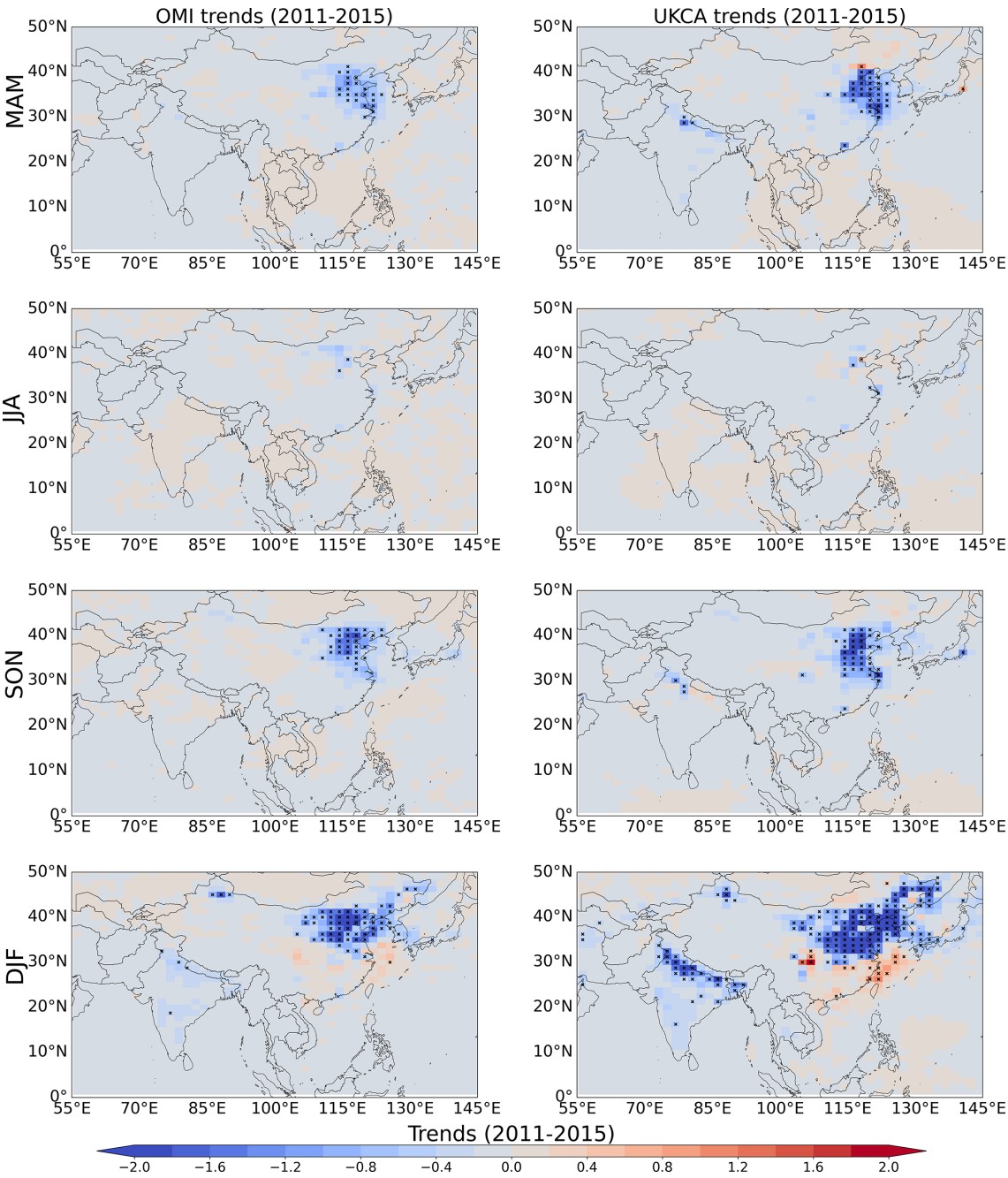

**Figure 10.** Trends of tropospheric column NO$_2$ ($10^{15}$ molec. cm$^{-2}$ yr$^{-1}$) from 2011 to 2015 from OMI (left) and UKCA (right) for the four seasons. Scatter plots of these data are shown in Fig. 12. Equivalent data in percentages are shown in Fig. S9. Crosses indicate grid squares with significant trends.

emissions has significant effects on the diurnal cycle of the simulated NO$_2$ column, tending to increase it during daylight hours as this is when more emissions occur. Hence, the inclusion of diurnally varying emissions would likely exacerbate the model–observation differences seen in this study.

Figures 6 and 7 show that the model is overestimating NO$_2$ column over the more polluted regions but underestimating it over the cleaner regions. This may reflect a lack of PAN for-

mation or equivalent sequestration of NO$_x$ in other reservoir species. PAN is a compound that locks up NO$_2$ in a reaction with the PA (peroxy acetyl) radical (Fiore et al., 2018). The PA comes from oxidation of certain volatile organic compounds (VOCs). PAN is stable at cold temperatures but unstable at high temperatures, decomposing back to NO$_2$ and PA. If PAN formation is too low (e.g. because VOCs are too

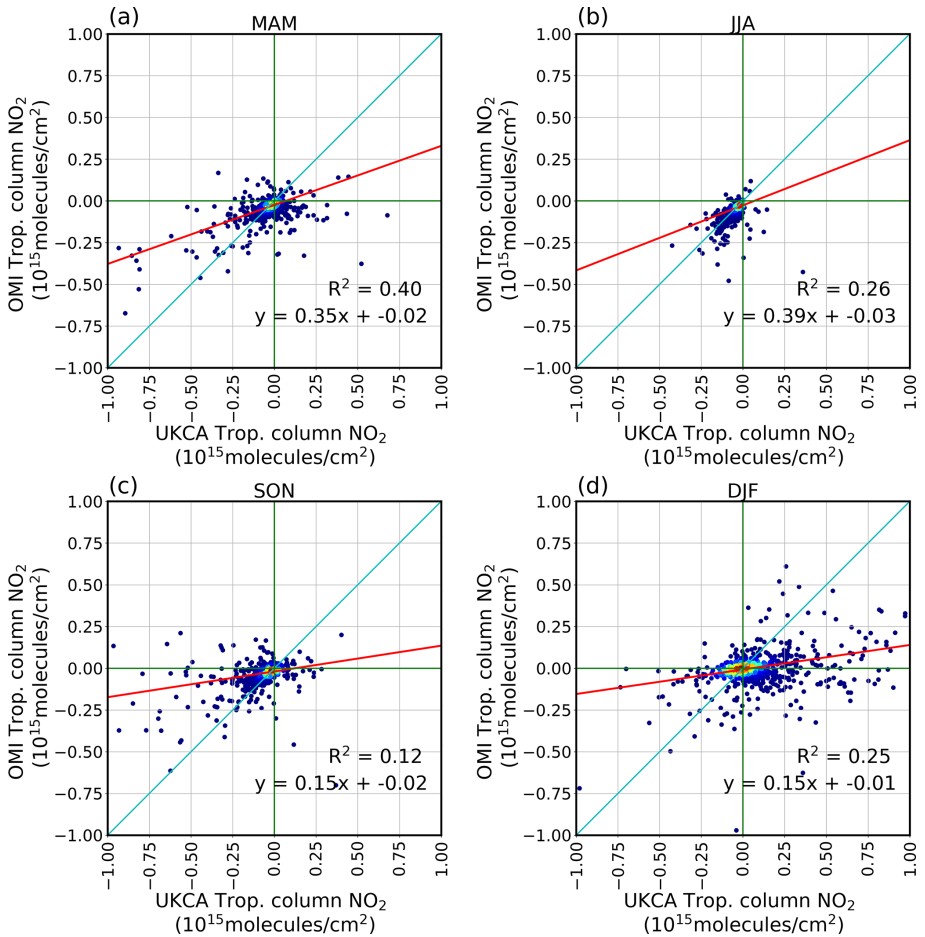

**Figure 11.** Scatter plot of UKCA and OMI tropospheric column NO$_2$ trends ($10^{15}$ molec. cm$^{-2}$ yr$^{-1}$) from 2005 to 2011 by season. The 1 : 1 line is shown in cyan colour and the best fit line in red colour. Data points shown as a heat map, where red corresponds to more data. The equations of the best fit and the coefficient of determination ($R^2$) are also shown. Note that $N = 2050$ data points are used for the fit, though some points fall outside the scales shown.

low), this may cause more NO$_2$ in source regions and less transport of NO$_2$ to remote regions.

Another potential contributing factor to the overestimation of NO$_2$ in source regions may be the underestimation of heterogeneous conversion of N$_2$O$_5$ to nitrate aerosol (e.g. Dentener and Crutzen, 1993; Riemer et al., 2003; Chen et al., 2018) in UKCA. These modelling studies have shown that this heterogeneous chemistry tends to reduce NO$_x$, especially during winter and in polluted regions with high aerosol loads, and it seems likely that the aerosol surface areas simulated by UKCA in these regions are underestimated.

Modelled trends in column NO$_2$ over S and E Asia are larger than trends seen in the OMI data, particularly during DJF. This is partly explained by the general overestimation of NO$_2$ columns, especially in polluted areas. Upward trends in aerosols would tend to enhance heterogeneous loss of oxidised N, so the underestimation of this process in UKCA would lead to an overestimate of NO$_2$ trends. Some of the model–observation discrepancies may also reflect un-

certainties in emissions magnitudes, spatial distributions, and trends.

Many of these reasons for model–observation differences are likely to be present in other models. Future research, such as a NO$_x$-focussed model intercomparison and evaluation (cf. van Noije et al., 2006) would help identify and quantify how widespread such problems may be amongst models. Given the central importance of NO$_x$ for multiple environmental issues investigated by models, such future research should be a high priority.

## 5 Conclusions

In this work, we evaluated tropospheric column NO$_2$ from the UKCA model using OMI satellite retrievals over S and E Asia. This required sampling the model at the satellite overpass time and application of vertical weighting profiles (averaging kernels) that account for how the satellite retrieval is influenced by the presence of clouds. UKCA can capture

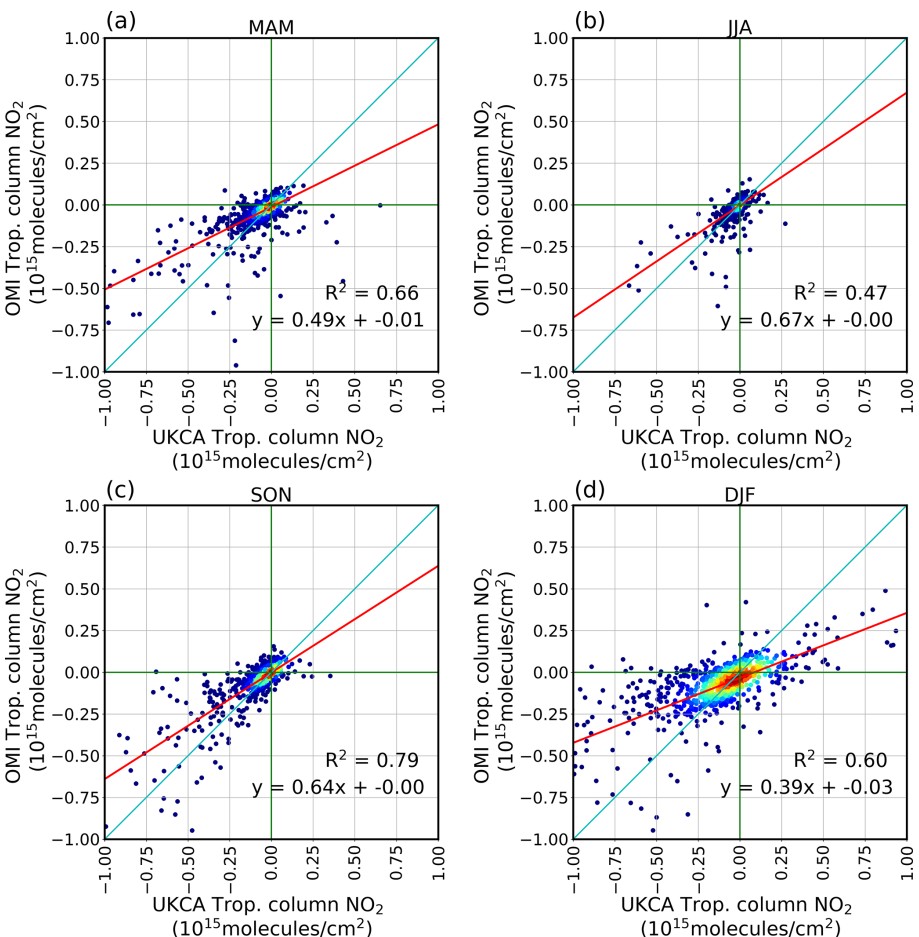

**Figure 12.** Scatter plots of UKCA and OMI tropospheric column NO$_2$ trends ($10^{15}$ molec. cm$^{-2}$ yr$^{-1}$) from 2011 to 2015 by season. The 1 : 1 line is shown in cyan colour and the best fit line in red colour. Data points shown as a heat map, where red corresponds to more data. The equations of the best fit and the coefficient of determination ($R^2$) are also shown. Note that $N = 2050$ data points are used for the fit, though some points fall outside the scales shown.

the NO$_2$ seasonality over S and E Asia but generally overestimates NO$_2$ column, especially in polluted regions during winter. UKCA overestimates column NO$_2$ near source regions but underestimates column NO$_2$ in remote regions, suggesting it is not converting enough NO$_2$ into longer-lived reservoir species such as PAN. Overestimations in polluted regions may be due to the UKCA model underestimating heterogeneous conversion of N$_2$O$_5$ to nitrate aerosol, which has been shown to quite strongly reduce NO$_x$ levels in the presence of aerosol, which is present at high levels across much of the region. UKCA also overestimates trends in the NO$_2$ column over the region. The underestimation of heterogeneous chemistry may be further contributing to the trend overestimations, as the influence of increases in aerosols over time will be missed.

Given the importance of accurate simulation of oxidised N for many processes important to the climate, air quality, and wider environment, further investigation of these discrepancies in simulated NO$_2$ in UKCA is required. In particular, we recommend the inclusion of schemes to more comprehensively represent heterogeneous chemistry and diurnal variation in emissions together with the exploration of the VOC emissions and PAN formation mechanisms in the model to see if their improved representation can lead to improvements in the simulation of column NO$_2$. We also recommend similar studies with other models to understand if these issues are common across models.

**Code and data availability.** This work used the United Kingdom Chemistry and Aerosol model. The model outputs were preprocessed using netCDF Operator (NCO) and Climate Data Operator (CDO). The analysis was carried out using Python. The UKCA model code is available at https://code.metoffice.gov.uk/trac/roses-u/browser/b/o/1/0/8/trunk (Met Office, 2025; log-in required). The supporting data is available at https://doi.org/10.7488/ds/7885 (Pandey and Stevenson, 2025).

**Supplement.** The supplement related to this article is available online at https://doi.org/10.5194/acp-25-1-2025-supplement.

**Author contributions.** AKP and DSS conceptualised and planned the research study. AKP performed the UKCA model simulations with support from DSS. AKP performed the model and satellite data analysis with the help of AZ. RJP and MPC helped in the satellite and model data comparison. KK and RH commented on the manuscript. AKP and DSS wrote most of the first draft. All authors helped to shape the paper content by editing prior versions of the paper.

**Competing interests.** The contact author has declared that none of the authors has any competing interests.

**Acknowledgements.** This work used the ARCHER and ARCHER2 UK National Supercomputing Service (http://www.archer.ac.uk/, last access: 16 January 2020, and https://www.archer2.ac.uk/, last access: 24 August 2023, respectively). Authors acknowledge ARCHER and ARCHER2 for supercomputing resources. The authors thank the United Kingdom Chemistry and Aerosol team, especially Luke Abraham for support in running the UKCA model and Paul Griffiths for helping us to understand the model's heterogeneous chemistry. We are grateful to Mohit Dalvi (Met Office) for providing the model dumps. This work used JASMIN, the UK's collaborative data analysis environment (http://jasmin.ac.uk, last access: 24 April 2025). We acknowledge the NASA Earth Science Division and KNMI for funding the OMI NO$_2$ development and the archiving of standard and DOMINO products via Tropospheric Emission Monitoring Internet Service (TEMIS – http://www.temis.nl/index.php, last access: 24 April 2025), respectively.

**Financial support.** Alok K. Pandey has been supported by the Commonwealth Rutherford Fellowship (Commonwealth Scholarship Commission grant no. INRF2017-196) and the Department of Science and Technology, India, and British Council UK Newton Bhabha Programme (grant no. DST/INSPIRE/NBHF2015/5). David S. Stevenson has been supported by the UKRI Global Challenges Research Fund South Asian Nitrogen Hub (grant no. NE/S009019/2). Funding for Richard J. Pope and Martyn P. Chipperfield has been provided by the UK NERC National Centre for Earth Observation (NCEO).

**Review statement.** This paper was edited by Tanja Schuck and reviewed by three anonymous referees.

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

## Remarks from the typesetter

**TS1** Since we have spotted these values were incorrect, it seems sensible to correct these for the final publication. The change makes no material difference to any of the paper's conclusions. We had originally incorrectly quoted ranges of values, based on the individual pixels within the box, but the text indicates they are total emissions from the box regions, so these should just be single values, not ranges. We have now calculated totals and would like to correct this sentence to read: "In 2015, total surface NO emissions were $2.83\,\mathrm{Tg\,N\,yr^{-1}}$ from the box over E China and $0.31\,\mathrm{Tg\,N\,yr^{-1}}$ from the N India box."