# Peer review of "Evaluating tropospheric nitrogen dioxide in UKCA using OMI satellite retrievals over South and East Asia"

_EGUsphere, 2024_

## Author Comment (AC1)

**Response to Reviewer 1**

Reviewer's comments are in italics, followed by our responses in non-italics.

*This paper compares $NO_2$ and its variability in the UKCA model over South and East Asia to OMI observations. The impact of sampling at the time of day corresponding to the satellite overpass and incorporating the OMI averaging kernel is investigated. The evaluation reveals areas and seasons of bias in the simulated $NO_2$ and suggests several possible factors that could contribute to the bias. The study provides a robust evaluation of the UKCA simulation of $NO_2$ over South and East Asia. The manuscript would benefit from additional discussion of whether the results can be generalized to other models or regions to broaden its scientific significance.*

This comment about generalizing is responded to below.

*General comments:*

*1. In the discussion, can you compare your results to other modeling studies or comment on whether the same biases are likely to affect other models? This could broaden the applicability of the study.*

We appreciate that being able to broaden the applicability of this study would be very useful. However, direct comparison to other modelling studies is difficult outside of something like a dedicated MIP (model intercomparison project), and, as far as the authors are aware, a large study like that has not been conducted since the early ACCENT model intercomparison (van Noije et al., 2006). That study mainly discussed differences between modelled and measured $NO_2$ in terms of discrepancies between $NO_2$ retrieval methods and the emissions used as inputs to the models rather than atmospheric processes within the individual models. Our study mainly describes the mechanics of how to meaningfully compare modelled and satellite measured $NO_2$ over S/E Asia, analyses recent trends, and discusses some potential reasons for model-measurement differences. A deeper understanding of these differences, firstly for the UKCA model, would be very useful. Extending the analysis to other models would also be very useful. However, our study is already fairly lengthy, and we feel that expanding it further is mainly beyond the scope of this particular paper. Nevertheless, we do extend the discussion to highlight that similar biases may well be present in other models, and further research, such as within a MIP dedicated to analysing $NO_x$ budgets within models and evaluating column $NO_2$, would be of great benefit to the wider improvement and assessment of atmospheric chemistry models.

*2. While the study focuses on South and East Asia, it would be helpful to include a figure or some discussion of how the model compares to OMI observations globally. Are the biases found over South and East Asia present in other regions as well? This might provide additional information on what the most likely cause is.*

We again agree that extending the analysis to look globally would be beneficial, however, we decided to focus our study on S/E Asia. Archibald et al. (2020; their Figures 18 & 19) shows a global comparison, for a similar model set-up. The reasons for focussing on S/E Asia are laid out in the manuscript: e.g., it is the region of the world with the largest $NO_2$ columns, the greatest pollution problems, and also has shown large and divergent trends in $NO_2$ over the period studied. All these reasons make S/E Asia an obvious initial target region. It would definitely be interesting to know if the biases we see are present elsewhere, and we agree this

may help pin down the causes. But we feel it is beyond our scope within this paper to extend the analysis globally; we hope to do this in future studies.

*3. Uncertainties in NOx emissions and emission trends are another potential cause of model biases. More discussion is needed on the NOx emissions used in the model and their uncertainties.*

We agree that emissions may cause some of the model biases and now include some discussion of this. Whilst it is possible that emissions in industrial regions are overestimated and remote emissions are underestimated, we think that a process bias (such as related to insufficient heterogeneous removal and/or PAN formation) is probably a more likely contributor to the model-measurement discrepancies that we see, but further investigation is needed to make stronger statements.

*Specific comments:*

*Page 5: Do you sample the model every day or only on days when there is a retrieval for a specific grid box? Is the model-obs comparison affected by clear-sky sampling?*

The UKCA model is sampled daily at the satellite overpass time (13:45 ± 15 minutes local time). For model-observation comparisons, only days and grid boxes with valid satellite retrievals are included. This ensures consistency in spatial and temporal sampling, reducing the potential for errors associated with inconsistent sampling. Although satellite data is filtered for clear-sky (i.e., cloud fraction < 0.2) conditions, the model is sampled under all conditions, including clouds, which may introduce bias. However, the UKCA model is nudged to ERA-Interim data, so when the real world has clear skies, the model should too, reducing transport and sampling inconsistencies and enhancing the reliability of the comparisons.

*Line 145: How were these regions chosen?*

Apart from whole country (India and China) analysis, we also selected $NO_x$ hotspot and cleaner regions to capture contrasting pollution profiles across South and East Asia. For instance, West China has surface $NO_x$ emission intensity that is approximately 30 times lower than East China, while South India's emissions intensity is about half that of North India (see new Figure S1, below). This approach allows for a comprehensive analysis across both high-emission and cleaner regions, providing broader insights into $NO_x$ distribution. We have added this clarification in the manuscript.

*Fig. S3: The caption of Fig. S3 says "trends" but the figure appears to show a timeseries.*

We have modified the caption of Figure S3 (Figure S4 of the revised manuscript).

*Figs. S5 and S6: Are these discussed in the text?*

Figures S5 and S6 (Figures S6 and S7 of the revised manuscript), which provide diurnal and seasonal insights into boundary layer height variations, have now been explicitly discussed in the manuscript. These figures are referred to in Sections 3.1 and 3.3 to explain how BLH dynamics influence $NO_2$ distribution and model performance.

*Line 195: Can you explain why the averaging kernel makes a larger difference in winter?*

In winter, the boundary layer height is lower, confining $NO_2$ closer to the surface within the planetary boundary layer (PBL). This trapping effect reduces the sensitivity of OMI, as the satellite struggles to detect $NO_2$ concentrated at lower altitudes. Consequently, the averaging kernels (AKs) have lower values near the surface, which potentially influence the comparisons between model and satellite data. When the dot product is taken between the AK and the model sub-column profile, the lowest sub-column contributions are substantially reduced, affecting the derived tropospheric column amount. Additionally, increased aerosol loads and shallow boundary layers prevalent in winter further limit the satellite's ability to observe $NO_2$ near the surface. These factors collectively enhance the influence of the AK in winter, where corrections for vertical sensitivity are most pronounced. Studies by Boersma et al. (2008, 2016) emphasize the importance of AKs in mitigating discrepancies between model and satellite data under such conditions, while Martin (2008) highlights their role in adjusting for seasonal and vertical variability in $NO_2$ profiles.

*Line 197: The statement "clearly demonstrates the importance of…" may need a caveat as the difference looks quite small in summer.*

We agree that the statement could benefit from clarification. While applying the averaging kernels (AKs) significantly reduces the winter bias (from ~80% to ~50%), the difference is indeed smaller in summer, with biases close to zero. We have modified the statement to reflect that the importance of the AKs varies seasonally, with a more pronounced effect in winter than in summer.

*Section 3.5: It would be helpful to plot a timeseries or trends for the NOx emissions for each of the regions.*

We have now included time series of NOx emissions intensity (emissions per unit area) for each region, as suggested (Figure S1).

*Section 3.5: How do the trend magnitudes compare in terms of % trends?*

We now include plots of % trends (Figures S8 and S9).

*Discussion: Could errors in the assumed emission trends also be a cause of model-obs mismatch? This should be discussed.*

Errors in the assumed emission trends may well also be part of the mismatch between model results and observations, and we now include this in our discussion. However, the overall upward then downward trend in emissions over China, and upward trend over India are also seen in the satellite data, suggesting the signs of trends in regional emissions are well represented. Modelled trends are larger than seen in the measurements, particularly in China in winter. Sensitivity experiments with reduced emissions trends would be required to quantitatively explore this as a cause of the mismatch. Such experiments would be useful but have not been performed in this study; we consider detailed sensitivity experimental modelling is beyond the scope of this study but would be very interesting for future research.

**References**

Archibald, A. T., O'Connor, F. M., Abraham, N. L., Archer-Nicholls, S., Chipperfield, M. P., Dalvi, M., Folberth, G. A., Dennison, F., Dhomse, S. S., Griffiths, P. T., Hardacre, C., Hewitt, A. J., Hill, R. S., Johnson, C. E., Keeble, J., Köhler, M. O., Morgenstern, O., Mulcahy, J. P., Ordóñez, C., Pope, R. J., Rumbold, S. T., Russo, M. R., Savage, N. H., Sellar, A., Stringer, M., Turnock, S. T., Wild, O., and Zeng, G.: Description and evaluation of the UKCA stratosphere–troposphere chemistry scheme (StratTrop vn 1.0) implemented in UKESM1, Geosci. Model Dev., 13, 1223–1266, https://doi.org/10.5194/gmd-13-1223-2020, 2020.

Boersma, K. F., Braak, R. and van der A, R. J.: Dutch OMI NO 2 (DOMINO) data product v2.0, Image (Rochester, N.Y.), 2(2), 1–21, 2011.Boersma, K. F., Jacob, D. J., Eskes, H. J., Pinder, R. W., Wang, J. and van der A, R. J.: Intercomparison of SCIAMACHY and OMI tropospheric NO2 columns: Observing the diurnal evolution of chemistry and emissions from space, J. Geophys. Res. Atmos., 113(16), 1–14, doi:10.1029/2007JD008816, 2008.

Boersma, K. F., Vinken, G. C. M., and Eskes, H. J.: Representativeness errors in comparing chemistry transport and chemistry climate models with satellite UV–Vis tropospheric column retrievals, Geosci. Model Dev., 9, 875–898, https://doi.org/10.5194/gmd-9-875-2016, 2016.

Martin, R. V.: Satellite remote sensing of surface air quality, Atmos. Environ., 42(34), 7823–7843, doi:10.1016/j.atmosenv.2008.07.018, 2008.

van Noije, T. P. C., Eskes, H. J., Dentener, F. J., Stevenson, D. S., Ellingsen, K., Schultz, M. G., Wild, O., Amann, M., Atherton, C. S., Bergmann, D. J., Bey, I., Boersma, K. F., Butler, T., Cofala, J., Drevet, J., Fiore, A. M., Gauss, M., Hauglustaine, D. A., Horowitz, L. W., Isaksen, I. S. A., Krol, M. C., Lamarque, J.-F., Lawrence, M. G., Martin, R. V., Montanaro, V., Müller, J.-F., Pitari, G., Prather, M. J., Pyle, J. A., Richter, A., Rodriguez, J. M., Savage, N. H., Strahan, S. E., Sudo, K., Szopa, S., and van Roozendael, M.: Multi-model ensemble simulations of tropospheric NO2 compared with GOME retrievals for the year 2000, Atmos. Chem. Phys., 6, 2943–2979, https://doi.org/10.5194/acp-6-2943-2006, 2006.

[Figure]

**New Figure S1** Monthly (2005-2015) NO surface emission intensity (g N month$^{-1}$ m$^{-2}$) over India and China, along with the polluted (North India and East China) and relatively clean (South India and West China) regions analyzed (see Figure 1c).

[Figure]

**New Figure S8** Percent trends of tropospheric column NO$_2$ from 2005 to 2011 from OMI (left) and UKCA (right) for the four seasons. Absolute trends are shown in Figure 9. Grid boxes with significant trends are indicated by crosses.

[Figure]

**New Figure S9** Percent trends of tropospheric column NO$_2$ from 2011 to 2015 from OMI (left) and UKCA (right) for the four seasons. Absolute trends are shown in Figure 10. Grid boxes with significant trends are indicated by crosses.

---

## Author Comment (AC2)

**Response to Reviewer 2**

Reviewer's comments are in italics, followed by our responses in non-italics.

*This paper provides valuable insights for modeling $NO_2$ in south and east Asia. A comprehensive comparison between UKCA modeled $NO_2$ concentrations and OMI $NO_2$ observations and trend analyses using both datasets are performed. The results show that in south and east Asia, UKCA tends to overestimate $NO_2$ compared to OMI while exhibits similar trends over 2005-2015. However, more detailed and quantitative discussions need to be done to better illustrate the results.*

*Major comments:*

*(1) To me, the motivation of this study remains unclear. Does this paper aim at model evaluation? If so, I recommend a review of CCMs in the introduction, and more technical details of UKCA in section 2.1. Also, more background information is needed to explain why this paper focuses on south/east Asia besides the large population, e.g., lack of ground monitoring networks, and discrepancies between model and observations.*

We agree we need to clarify the paper's motivation. Its motivations include: (i) to describe some of the difficulties in comparing column $NO_2$ in an atmospheric chemistry model with satellite measurements; (ii) to present such a comparison using the UKCA model; and (iii) to speculate on some of the atmospheric chemical processes that may contribute to model-measurement differences.

We take the point that we should clarify if what we are presenting is a model evaluation. We attempt to compare, as best we can, modelled $NO_2$ columns (and trends) with those measured by OMI, to try and draw conclusions about how well the model is performing. We feel that could be described as model evaluation. However, it is only a partial evaluation of how well the model is representing $NO_2$: e.g., we do not additionally compare with other available measurements of $NO_2$ (e.g., from surface sites and aircraft campaigns), nor associated relevant measurements (e.g., $NO_2$ deposition fluxes, or photolysis rates, etc.). A more comprehensive assessment of oxidised nitrogen would be very useful, but we feel is beyond the scope of the present study.

Previous studies have presented comparisons between modelled and satellite column $NO_2$ and generally carry out these comparisons using sensible methodologies (i.e. accounting for averaging kernels and time of satellite overpass), however we have not previously seen a simple step-by-step explanation of such a methodology and why it is important.

Understanding what a satellite measures is crucial for evaluating modelled $NO_2$ and seeing why that model evaluation carries greater uncertainties at certain locations and times of year (i.e. at higher latitudes in winter).

The paper focuses on S/E Asia as this is where satellites measure the largest $NO_2$ columns associated with anthropogenic emissions, and there have been large trends in these emissions over recent years.

We now include a brief description of the representation of $NO_2$ in UKCA in the introduction and the model description section. It is difficult to generalise this to all CCMs as they represent atmospheric chemistry to different degrees, but the $NO_2$ chemistry in UKCA is probably typical.

*(2) Overall, there is a lack of comparisons with previous studies, e.g., is overestimating NOx a common problem for CCMs; does this study improve any existing problems? Also, the role and uncertainties of nitrogen deposition should be included. Moreover, the discussion and*

*conclusions need to be more quantitative. The author mentions uncertainties in NOx emission inventories and the representation of heterogeneous chemistry without quantifying the uncertainties, as a result, it is difficult to draw any strong conclusions.*

Archibald et al. (2020) briefly evaluate $NO_2$ columns in UKCA by comparing with OMI data. These authors use a similar version of the model to that used in our study. E.g., Figure 18 of Archibald et al. (2020) shows global maps of simulated 2005-2014 DJF and JJA tropospheric $NO_2$ columns, together with UKCA-OMI differences. These results highlight similar issues to those found in our study: $NO_2$ is overestimated in UKCA over polluted regions of Asia, particularly during winter. Averaged over the whole latitude band 30-60N, the model overestimates DJF $NO_2$ and underestimates JJA.

We agree it is difficult to draw strong conclusions as we have not performed any sensitivity experiments (e.g., changing emissions or varying model chemistry). We feel this is for follow-up studies; our study only attempts to document current model performance and speculate on potential reasons for model discrepancies.

*(3) Trend analysis:*
*What is the statistical method used for trend analysis? Are annual trends deseasonalized? Please add a paragraph of methodology description in Section 2.*
*What are the statistical significance values? I suspect that a lot of regions do not have statistically significant trends.*

We applied linear regression to calculate trends in $NO_2$ concentrations. The annual means of $NO_2$ were used for the analysis, which inherently removes seasonal variability, ensuring that the trends reflect long-term changes rather than seasonal fluctuations.
Statistical significance was tested at a 95% confidence level ($\alpha$=0.05). The t-statistic for each trend was calculated as t=trend/SE, where SE is the standard error of the trend estimate. The critical t-value ($t_{critical}$) was obtained from the t-distribution using degrees of freedom (df=n-2, where n is the number of years). Trends were considered significant if $|t|>t_{critical}$, indicating that the trend is unlikely to have occurred by chance with 95% confidence. Grid-boxes with significant trends are indicated by crosses on the revised Figures 9 and 10 (below).
Not all regions exhibit statistically significant trends. We have added a paragraph in Section 2 detailing the methodology for trend analysis, including the significance testing procedure to enhance clarity.

*Is 2011 double-counted in both 2005-2011 and 2011-2015 periods?*

Data from 2011 is included in both the time periods analysed. We wouldn't describe this as double-counting, as this makes it sound like a mistake. The year 2011 is deliberately included in both periods.

*(4) Please highlight the novelty of this study.*

Nitrogen dioxide columns in the UKCA model have only been briefly evaluated previously (Archibald et al., 2020). This study extends that analysis, looking in detail at diurnal and seasonal variations, and trends over 2005-2015. This study highlights significant discrepancies between UKCA and OMI measurements, especially in winter over China, and suggests some reasons why the model is under-performing. We hope this provides a useful guide for future model development and further evaluation of $NO_x$, both in UKCA and potentially in other

models. These model discrepancies are important to address, since $NO_x$ is important for a wide range of environmental problems that models are used to investigate.

*Minor comments:*

*Line 52 and line 296: NOx: subscript*
Modified

*Line 82: BLH affects $NO_2$ column surface $NO_2$*
Modified

*Figure 5: it would be better to show the 2005 – 2015 mean instead of just using 1-year data. Also, please plot the number of OMI NO2 pixels for each month.*

Thank you for your suggestion regarding Figure 5. We have revised the figure using the mean values for the years 2005–2015, as recommended (see below). However, regarding the number of OMI $NO_2$ pixels for each month, we encountered challenges in obtaining this data. The OMI pixels are irregular and follow a swath with varying sizes, making it difficult to compile consistent data over the ten years. The uncertainty bars partly reflect these variations in sampling, and we are confident that there are sufficient pixels for each month to allow us to perform a useful model-measurement comparison.

*Figure 7 (a): please adjust the legend to avoid blocking the text*

We have modified the legend in Figure 7(a) to prevent it from blocking the text. This adjustment enhances the overall readability of the figure.

*Figure 11 and 12: add number of points to the scatter plot*

We have modified the figure captions to explicitly state that there are 2050 points used for each plot fit, while also acknowledging that some points are not on these plots as they fall outside the scales shown.

[Figure]

**Revised Figure 5** Comparison of monthly mean tropospheric column NO₂, averaged over 2005-2015 for the whole S/E Asia region (Figure 1) from OMI (green, with uncertainty indicated by the bars), and from UKCA sampled in three different ways: (i) simple monthly mean (blue); (ii) sampled at the OMI overpass time (cyan); and (iii) sampled at the overpass time and with satellite averaging kernels applied (magenta).

[Figure]

**Revised Figure 7** Scatter plots of OMI and UKCA tropospheric column NO$_2$ for the four seasons averaged over 2005-2015. Scatter data points are plotted as a heat map where red corresponds to more data. The 1:1 line is shown in cyan colour, best fit in red line (all data) and green line (lowest 90% of data). The equations of best fit and the coefficients of determination ($R^2$) are also shown in the respective colours.

[Figure]

**Revised Figure 9** Trends of tropospheric column $NO_2$ ($10^{15}$ molecules/cm$^2$/yr) from 2005 to 2011 from OMI (left) and UKCA (right) for the four seasons. Significant trends are indicated by crosses. Scatter plots of these data are shown in Figure 11. Equivalent data expressed as percentages are shown in Figure S8.

[Figure]

**Revised Figure 10** Trends of tropospheric column $NO_2$ ($10^{15}$ molecules/cm$^2$/yr) from 2011 to 2015 from OMI (left) and UKCA (right) for the four seasons. Significant trends are indicated by crosses. Scatter plots of these data are shown in Figure 12. Equivalent data expressed as percentages are shown in Figure S9.

---

## Author Comment (AC3)

**Response to Reviewer 3**

Reviewer's comments are in italics, followed by our responses in non-italics.

*This paper presents an evaluation/comparison of modeled tropospheric nitrogen dioxide in UKCA with OMI satellite observation. The model results are presented and compared with satellite data. The following are my comments;*

*Major comments:*
1. *The manuscript keeps mentioning evaluation/comparison of two datasets. For example, the title describes it as 'evaluation' but abstract section mentioned it as 'comparison'. From my understanding, comparison involves analyzing the similarities or differences while evaluation is a quantitative assessment of how well a model replicates reality (OMI satellite retrievals in this case). The authors need to clearly describe whether they are evaluating or comparing the two datasets.*

We apologize for any lack of clarity. As described in an earlier response to Reviewer 2, we consider our study a quantitative evaluation of tropospheric $NO_2$ columns (and their trends) over S/E Asia in the UKCA model.

2. *The need for the comparison is not well established in the introduction section. Is this the first-time comparison between UKCA model and OMI or the comparisons reported before for two datasets? Further, why did the authors choose OMI for evaluating/comparing the UKCA results? Are the OMI $NO_2$ observations evaluated before in the study region? and are good enough to make comparisons. If so, authors can mention some studies conducted for evaluation of OMI $NO_2$ columns in different regions.*

This study represents a comprehensive evaluation of the UKCA model using OMI $NO_2$ data over South and East Asia, a region with diverse and rapidly changing emission sources. OMI and UKCA $NO_2$ columns have previously been compared in Archibald et al. (2020). However, our current study goes into more detail about the comparison, considers other seasons, diurnal variations, and also temporal trends.

OMI was chosen for its long-term (since 2004), daily, high-resolution measurements of tropospheric $NO_2$, which provide robust datasets for evaluating model performance. The suitability of OMI observations for model comparison has been demonstrated in numerous studies. For example, Lamsal et al. (2014) evaluated OMI $NO_2$ data against in situ and surface-based observations over the continental U.S., confirming its reliability for capturing regional $NO_2$ distributions. Additionally, comparisons between OMI and ground-based MAX-DOAS observations in East Asia have demonstrated consistency (Irie et al. 2009), further supporting OMI's suitability for model evaluation in this region.

We have revised the introduction to include these references and clarify the motivation.

3. *The paper is describing the diurnal simulations from UKCA model, but the OMI only provides one measurement per day (as described by the authors), then what about evaluation/comparison of diurnal variation? or the authors only evaluating for satellite overpass time? If this is the case, then figure 02 may lead to a confusion for the readers as if diurnal variations are evaluated.*

We acknowledge that the OMI satellite provides only one measurement per day at its overpass time (~13:45 local time), which limits its ability to capture the full diurnal variation of $NO_2$. In our study, the comparison between the UKCA model and OMI observations is restricted to the satellite overpass time, where the model is sampled to match the temporal resolution of the satellite.

The diurnal variations shown in Figure 2 are derived solely from the UKCA model simulations and are not directly evaluated against OMI data. The purpose of presenting these diurnal profiles is to provide context for understanding how $NO_2$ concentrations vary throughout the day in the model, influenced by meteorological processes such as boundary layer dynamics and photochemistry. This helps illustrate the broader behaviour of $NO_2$ beyond the satellite overpass time.

It is important to appreciate the large diurnal cycle of $NO_2$ and the seasonal variation in this diurnal cycle, as this affects uncertainties in the model-satellite comparison. For example, as we state in the manuscript, during summer over China the satellite retrieval time is during a broad minimum, whereas during winter the retrieval time is during a time of day when $NO_2$ column is changing. This means that the wintertime comparisons carry larger uncertainties. Without illustrating the diurnal cycle this would not be obvious.

To address potential confusion, we have revised the caption for Figure 2 to clarify that the diurnal variations are simulated by the model and not directly compared with OMI observations. Additionally, we ensure that the text explicitly states that the model-observation comparisons are limited to the satellite overpass time.

*Minor Comments*

1. *Line 99-100: "The model's horizontal resolution (N96: 1.875° longitude × 1.25° latitude) is much coarser than the satellite data products used." I think there is a need to explain how the data is matched on a spatial scale for fair evaluation/comparison.*

The model-satellite comparison requires careful spatial matching to ensure fair evaluation. Although the UKCA model has a coarser horizontal resolution (N96: 1.875° longitude × 1.25° latitude) compared to the higher resolution of OMI satellite products, we accounted for this difference by interpolating the satellite data to the model grid. Specifically, the satellite-retrieved $NO_2$ data were averaged over spatial areas corresponding to the UKCA grid cells, ensuring consistency between the two datasets. This approach allows for a meaningful comparison while minimizing biases introduced by the resolution mismatch.

2. *Line 109: What's the resolution of ECMWF ERA-interim data used here?*

The ECMWF ERA-Interim data used here is obtained at T255 (78 km) resolution on ECMWF hybrid-p levels, provided at six-hourly intervals. These variables are then interpolated to the N96 resolution. The resolution information has been added in the revised manuscript.

3. *Line 144: The study area is mentioned at the end of methodological section. I would suggest moving the description of study area at the start of methodological section.*

We agree that moving the study area description to the start of the methodological section would improve the flow and clarity. We have adjusted the structure accordingly to introduce the study area at the beginning of Section 2.

> 4. *For Figure 1, why the percentage change is much higher over the oceans, what could be the possible sources/reasons? also it would be helpful to show the emissions for 2005.*

The higher percentage changes observed over the oceans are primarily due to large changes in shipping emissions. As global shipping activities have increased, they have contributed significantly to emissions over ocean areas. Additionally, we have modified Figure 1 to include the emissions data for 2005, as suggested.

> 5. *Line 174-175: Other than the pollution, what could be the role of meteorological parameters in modulating the vertical distribution/height of $NO_2$ in different regions of study area?*

Thanks for this question. Meteorological factors, particularly temperature, play a significant role in modulating the vertical distribution and lifetime of $NO_2$ (Atkinson, 2000; Liu et al., 2016). Lower winter temperatures slow chemical reactions, extending the atmospheric lifetime of $NO_2$. This effect, combined with shallow boundary layers and stable atmospheric conditions, contributes to higher $NO_2$ concentrations near the surface and alters its vertical distribution.

**References**

Archibald, A. T., O'Connor, F. M., Abraham, N. L., Archer-Nicholls, S., Chipperfield, M. P., Dalvi, M., Folberth, G. A., Dennison, F., Dhomse, S. S., Griffiths, P. T., Hardacre, C., Hewitt, A. J., Hill, R. S., Johnson, C. E., Keeble, J., Köhler, M. O., Morgenstern, O., Mulcahy, J. P., Ordóñez, C., Pope, R. J., Rumbold, S. T., Russo, M. R., Savage, N. H., Sellar, A., Stringer, M., Turnock, S. T., Wild, O., and Zeng, G.: Description and evaluation of the UKCA stratosphere–troposphere chemistry scheme (StratTrop vn 1.0) implemented in UKESM1, Geosci. Model Dev., 13, 1223–1266, https://doi.org/10.5194/gmd-13-1223-2020, 2020.

Atkinson, R. (2000). Atmospheric chemistry of VOCs and NOx. Atmospheric Environment, 34(12-14), 2063–2101. https://doi.org/10.1016/S1352-2310(99)00460-4

Irie, H., Kanaya, Y., Takashima, H., Gleason, J. F. and Wang, Z.: Characterization of OMI tropospheric $NO_2$ measurements in East Asia based on a robust validation comparison, Sci. Online Lett. Atmos., 5(1), 117–120, doi:10.2151/sola.2009-030, 2009.

Lamsal, L. N., Krotkov, N. A., Celarier, E. A., Swartz, W. H., Pickering, K. E., Bucsela, E. J., Gleason, J. F., Martin, R. V., Philip, S., Irie, H., Cede, A., Herman, J., Weinheimer, A., Szykman, J. J., and Knepp, T. N.: Evaluation of OMI operational standard $NO_2$ column retrievals using in situ and surface-based $NO_2$ observations, Atmos. Chem. Phys., 14, 11587–11609, https://doi.org/10.5194/acp-14-11587-2014, 2014.

Liu, F., Beirle, S., Zhang, Q., et al. (2016). NOx lifetimes and emissions of cities and power plants in polluted backgrounds estimated by satellite observations. Atmospheric Chemistry and Physics, 16(8), 5283–5298. https://doi.org/10.5194/acp-16-5283-2016

[Figure]

**Revised Figure 1** Surface nitrogen oxide (NO) emissions over S/E Asia (Tg N yr$^{-1}$) in (a) 2005 and (b) 2015; (c) Percentage change in the NO surface emissions from 2005 to 2015; (d) trends of NO surface emissions (Tg N month$^{-1}$) from 2005 to 2015 over India and China. Boxes shown in (b) indicate regions referred to in the text.

---

## Author Response (AR2)

**Responses to comments on revised manuscript (EGUSPHERE-2024-2686: Evaluating tropospheric nitrogen dioxide in UKCA using OMI satellite retrievals over South and East Asia, by A.K. Pandey et al.)**

Author responses in red below.

Public justification (visible to the public if the article is accepted and published):

Dear author team,

thanks for submitting the revised version of your study on NO2 over Asia. Unfortunately I could not track all responses to the referee comments in the revised manuscript. Please check for the points listed below and clarify which changes to the text were made.

Please cite all these changes to the manuscript in your response and refer to the positioning of changes with line numbers from manuscript version 3.
* * *
Response to comments by reviewer#1
* * *
-- general comment 1: in your response you state, that the discussion was extended with regard to model biases. Please clarify which changes in the manuscript were made regarding this detail.

*[General comment 1 in initial review: 1. In the discussion, can you compare your results to other modeling studies or comment on whether the same biases are likely to affect other models? This could broaden the applicability of the study.]*

Our response states that we can't directly compare our results to other modelling studies without something like a dedicated model intercomparison project, such as that presented by van Noije et al. (2006). We have extended the discussion to reflect this response:

Lines 339-344 note that model-observation discrepancies may reflect various uncertainties in emissions, and that to compare our results to other models, a new NOx-focussed model intercomparison and evaluation would be useful. We also highlight this in the paper's closing statement (lines 360-361).

-- question 3 on uncertainties:

*[General comment 3 in initial review: 3. Uncertainties in NOx emissions and emission trends are another potential cause of model biases. More discussion is needed on the NOx emissions used in the model and their uncertainties.]*

Lines 297-299 discuss some likely problems with the spatial distribution of emissions used.

Lines 194-196 and 320-324 discuss the neglect of the diurnal cycle of emissions.

The above discussions were already in the original submission. We added lines 339-340 to note that emissions may also contribute to the model-observation discrepancies. In our response we note that our paper focusses on potential process biases rather than emissions biases, hence the relative lack of discussion about emissions.

-- choice of regions

Lines 93-101 in the revised text present our justification for the choice of regions.

-- Discussion on errors in emission trends:

*[Comment in initial review: Could errors in the assumed emission trends also be a cause of model-obs mismatch? This should be discussed.]*

We added lines 339-340 to note that uncertainties in emissions' trends may also contribute to the model-observation discrepancies.

Any changes to the text done for these aspects? Please specify.
* * *
Response to comments by reviewer#2
* * *
-- (1) motivation

-- (4) novelty of study

I agree with the given responses to reviewer #2, but were there any changes to the text highlighting the motivations and the novelty of the study in the manuscript?

We substantially re-organised the Introduction (lines 30-85) to clarify the motivation and novelty of the study. The changes are summarised in our response to reviewer #2, and in the revised introduction text.